# Losses Can Be Blessings: Routing Self-Supervised Speech Representations Towards Efficient Multilingual and Multitask Speech Processing

**Yonggan Fu[1], Yang Zhang[2], Kaizhi Qian[2], Zhifan Ye[3], Zhongzhi Yu[1]**
**Cheng-I Lai[4], Yingyan (Celine) Lin[1]**
[1]Georgia Institute of Technology, [2]MIT-IBM Watson AI Lab, [3]Rice University, [4]MIT CSAIL
`{yfu314,zyu401,celine.lin}@gatech.edu`
`{yang.zhang2,kqian}@ibm.com {zy50}@rice.edu {clai24}@mit.edu`

## Abstract

Self-supervised learning (SSL) for rich speech representations has achieved empirical success in low-resource Automatic Speech Recognition (ASR) and other speech processing tasks, which can mitigate the necessity of a large amount of transcribed speech and thus has driven a growing demand for on-device ASR and other speech processing. However, advanced speech SSL models have become increasingly large, which contradicts the limited on-device resources. This gap could be more severe in multilingual/multitask scenarios requiring simultaneously recognizing multiple languages or executing multiple speech processing tasks. Additionally, strongly overparameterized speech SSL models tend to suffer from overfitting when being finetuned on low-resource speech corpus. This work aims to enhance the practical usage of speech SSL models towards a win-win in both enhanced efficiency and alleviated overfitting via our proposed $S^3$-Router framework, which for the first time discovers that simply discarding no more than 10% of model weights via only finetuning model *connections* of speech SSL models can achieve better accuracy over standard weight finetuning on downstream speech processing tasks. More importantly, $S^3$-Router can serve as an all-in-one technique to enable (1) a new finetuning scheme, (2) an efficient multilingual/multitask solution, (3) a state-of-the-art ASR pruning technique, and (4) a new tool to quantitatively analyze the learned speech representation. We believe $S^3$-Router has provided a new perspective for practical deployment of speech SSL models. Our codes are available at: `https://github.com/GATECH-EIC/S3-Router`.

## 1 Introduction

Deep neural network (DNN) breakthroughs have tremendously advanced the field of Automatic Speech Recognition (ASR). However, one major driving force for powerful DNNs, i.e., the availability of a large amount of training data, is not always possible for ASR. This is because collecting large-scale transcriptions is costly, especially for low-resource spoken languages around the world, limiting the wide application of deep ASR models. Fortunately, recent advances in self-supervised learning (SSL) for rich speech representations [1, 2, 3, 4, 5, 6, 7, 8, 9] have achieved empirical success in low-resource ASR, where SSL models pretrained on raw audio data without transcriptions can be finetuned on low-resource transcribed speech to match the accuracy of their supervised counterparts.

However, there exists a dilemma between the trends of speech SSL models and the growing demand for speech processing applications on the edge. While advanced speech SSL models become increasingly larger to learn more generalizable features, it is highly desired to process the captured speech

36th Conference on Neural Information Processing Systems (NeurIPS 2022).

signals in real time on edge devices, which have limited resources and conflict with the prohibitive complexity of existing speech SSL models. Such an efficiency concern would be more severe in multilingual/multitask scenarios where simultaneously recognizing multiple languages or executing multiple speech processing tasks is required: if one separate model is finetuned for each target language/task, the storage and computational cost will be significantly increased, thus prohibiting the practical deployment of existing speech SSL models. Additionally, strongly overparameterized speech SSL models tend to suffer from overfitting when being finetuned on a low-resource speech corpus [10, 11, 12, 13, 14], limiting the achievable accuracy improvement brought about by more parameters and thus the achievable performance-efficiency trade-off.

In this work, we aim to facilitate the practical usage of speech SSL models towards a win-win in enhanced efficiency and alleviated overfitting for boosting task accuracy under low-resource settings. Excitingly, we develop a framework, dubbed **S**elf-**S**upervised **S**peech Representation **Router** ($\text{S}^3$-Router), that can serve as an all-in-one technique to tackle the aforementioned challenges and largely enhance the practical usage of speech SSL models, contributing (1) a new finetuning scheme for downstream speech processing, i.e., finetuning the connections of the model structure via learning a binary mask on top of pretrained model weights, which notably alleviates model overfitting and thus improves the achievable accuracy over standard weight finetuning methods under a low-resource setting; (2) a multilingual/multitask technique via learning language-/task-specific binary masks on top of shared model weights inherited from SSL pretraining; (3) a competitive ASR pruning technique to trim down the complexity of speech SSL models while maintaining task accuracy; and (4) a new tool to quantitatively analyze what is encoded in speech SSL models thanks to the learned masks' binary nature on top of shared model weights. We summarize our contributions below:

- We propose a framework dubbed $\text{S}^3$-Router, which offers an alternative to the mainstream finetuning of model *weights* that finetunes the *structure* of speech SSL models, by learning a binary mask on top of the pretrained model weights and integrating it with a novel mask initialization strategy customized for the pretrain-finetune paradigm;
- We are the first to discover that discarding no more than 10% of weights *without* finetuning pretrained model weights can achieve better task performance as compared to the mainstream method of weight finetuning on downstream speech processing tasks. Notably, our method can scale well to even larger models;
- We extend our $\text{S}^3$-Router framework for enhanced deployment efficiency, contributing (1) a novel multilingual/multitask solution, and (2) a competitive pruning technique achieving better performance-efficiency trade-offs than state-of-the-art (SOTA) ASR pruning techniques;
- We demonstrate the capability of $\text{S}^3$-Router as a tool to quantitatively understand what is encoded in speech SSL models thanks to the binary nature of the learned masks.

$\text{S}^3$-Router has opened up a new perspective for empowering efficient multilingual/multitask speech processing and enhancing our understanding about what is encoded in speech SSL models.

## 2   Related Work

**Automatic speech recognition.** Early ASR systems [15, 16, 17, 18, 19, 20] were mainly based on the combinations of hidden Markov models (HMM) with Gaussian mixture models or DNNs, and often contain multiple modules (e.g., an acoustic model, a language model, and a lexicon model) trained separately. Recent works process raw audio sequences end-to-end, including CTC [21]-based models [22, 23, 24, 25, 26], recurrent neural network(RNN)-transducers [27, 28, 29, 30], sequence-to-sequence models [31, 32, 33, 34, 35, 36], and transformer-based models [37, 38, 39]. Specifically, transformer-based models have been widely adopted as speech SSL models [6, 9, 7].

**Self-supervised learning for speech representation.** Considering the high cost of collecting large-scale transcriptions, learning rich speech representations via SSL has become crucial and promising for empowering low-resource ASR. Early works [40, 41, 42, 43, 44, 45, 46, 47] build generative models for speech with latent variables. Recently, prediction-based SSL methods have become increasingly popular, where the models are trained to reconstruct the contents of unseen frames [48, 49, 50, 51, 52, 53, 1, 2] or contrast the features of masked frames with those of randomly sampled ones [3, 4, 5, 6, 7, 8, 9]. We refer the readers to a recent survey [54] for more details. Among prior arts, [55] is a pioneering work relevant to our method, and adopts masking as an alternative to weight finetuning on pretrained language models for natural language processing (NLP).

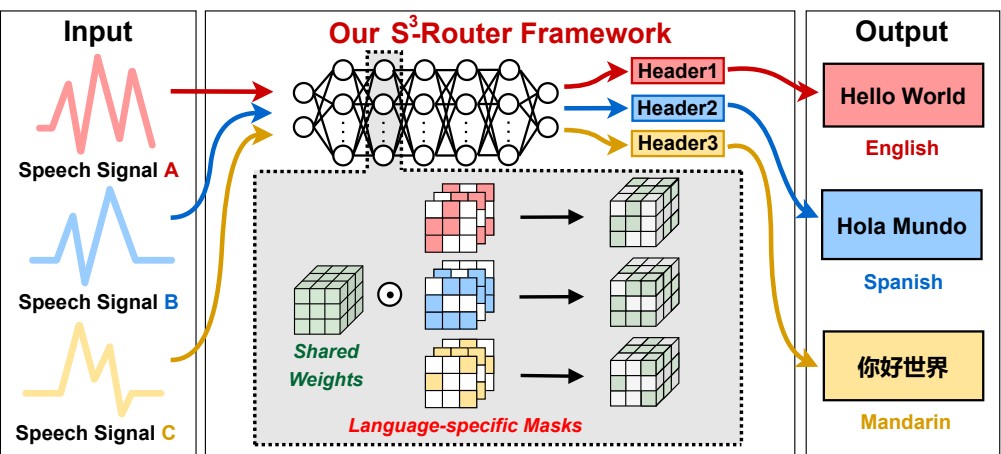

Figure 1: An overview of our S³-Router framework, which receives multilingual speech signals denoted as A, B, and C here and then outputs the corresponding text transcript of predication, based on one *shared weight* model together with language-/task-specific *binary* masks.

Nevertheless, our S³-Router is non-trivially different from [55] in that (1) we target SSL models in the speech domain and the learned speech representation is required to be generalizable across different spoken languages under a cross-lingual transfer setting, where the speech SSL models suffer from a higher risk of overfitting on downstream low-resource tasks as compared to NLP, making our findings non-trivial contributions; (2) S³-Router features a win-win in both efficiency and accuracy thanks to our proposed mask initialization strategy, which plays a crucial role in maintaining task accuracy under a high sparsity and enables the extension of S³-Router towards a competitive pruning technique with SOTA accuracy; and (3) we further develop S³-Router for multilingual/multitask speech processing and as a simple yet effective tool for analyzing language-wise similarities.

**Multitask learning for speech processing.** There exists a growing demand for DNNs to simultaneously process multiple tasks [56, 57]. In the speech domain, previous works attempt to train a single model to solve multiple tasks [58, 59, 60, 61, 62, 63] or use an auxiliary task to enhance the accuracy of a primary task [64, 65, 66, 67, 68, 69]. Motivated by the success of SSL in low-resource downstream tasks, multitask/multilingual learning has been adopted in both SSL pretraining [70, 71, 72, 73, 7] to enrich learned speech representations, and finetuning scenarios [74]. For example, a recent relevant work [73] adopts language-adaptive pretraining on top of [7], where different sparse sub-networks are activated for different languages during multilingual pretraining and the gradients are accumulated on the shared super-network to learn better multilingual representations. In contrast, S³-Router is fundamentally different, as it targets the finetuning stage of a given pretrained speech SSL model and only tunes the connections of the model structure *without* tuning the SSL pretrained weights.

**ASR pruning.** As ASR models become increasingly overparameterized for abstracting generalizable representations, pruning large-scale ASR models has drawn a growing attention. Early works trim down either the decoding search spaces [75, 76, 77, 78, 79, 80] or HMM state space [81]. Modern ASR paradigm has gradually shifted its focus to pruning end-to-end ASR models [82, 83, 84, 85, 86, 87, 88, 89, 90, 91, 92, 93]. Recently, [94, 95, 96] prune speech SSL models towards more efficient low-resource ASR, e.g., PARP [94] proposes a finetuning pipeline to identify the existence of lottery tickets within speech SSL models. In addition to the boosted pruning efficiency over PARP [94], S³-Router emphasizes the role of sparsity in encoding language-/task-specific information which can empower multitask/multilingual speech processing *without* tuning the pretrained weights.

## 3 The Proposed S³-Router Framework

### 3.1 Drawn Inspirations from Previous Work

Recent works [97, 98, 99] find that sub-networks featuring a decent inborn accuracy and adversarial robustness are hidden within randomly initialized networks *without* any weight training. Specifically, [97, 98] show that sub-networks with a decent accuracy, even matching that of their dense networks, can be identified from randomly initialized networks, and [99] shows an even stronger

evidence: merely updating the sparsity patterns of model connections *without* modifying the randomly initiated model weights can produce both accurate and adversarially robust models. These pioneering works imply that tuning the connections of the model structure, which can be characterized by the learned connection sparsity patterns, can be as effective as training the model weights. We hypothesize that **model sparsity not only can favor model efficiency, but also can serve as a similar role, i.e., another optimization knob, as model weights to encode language-/task-specific information**.

### 3.2 Formulation and Optimization of S³-Router

Inspired by the aforementioned intriguing hypothesis, we propose the S³-Router framework to empower efficient multilingual and multitask speech processing on top of SSL speech representations.

**Overview.** As shown in Fig. 1, given the raw audios of different spoken languages (or different tasks), S³-Router finetunes the connection patterns of the model structure for *each* target spoken language/task via optimizing language-/task-specific *binary masks* on top of the *shared weights* of a given speech SSL model, instead of finetuning the model weights as adopted in the common pretrain-finetune paradigm. Specifically, the learned binary masks of each language/task are multiplied with the shared model weights to mask out some connections within the given speech SSL model, where the remaining connections (or the induced connection sparsity) encode language-/task-specific information for different spoken languages and downstream tasks. Note that for each language or task, only one set of binary masks and one lightweight header, e.g., the classification head for ASR which is naturally non-shareable across languages due to diverse dictionary sizes, need to be independently trained, incurring negligible overhead (e.g.,$\leq$6.3% storage of the whole backbone model).

**Formulation.** Formally, our S³-Router framework can be formulated as:

$$\arg\min_{m_t} \sum_{(x_t,y_t)\in D_t} \ell_t(f(m_t \odot \theta_{SSL}, x_t), y_t) \quad s.t. \ ||m_t||_0 \leqslant k_t \tag{1}$$

where $(x_t, y_t)$ are the input audio and corresponding transcriptions/labels of a spoken language/task $t$ in a downstream dataset $D_t$, and $\theta_{SSL}$ is the SSL pretrained weights of the given speech SSL model $f$. Specifically, the mask set $m_t$ is applied on top of the model weights $\theta_{SSL}$, and optimized to minimize the loss function $l_t$, e.g., a CTC loss [21] for ASR, subject to an $L_0$ sparsity constraint, where the number of non-zero elements in $m_t$ is limited to $k_t$, which serves as a hyperparameter for controlling and balancing (1) the amount of language-/task-specific information encoded in the connection sparsity and (2) model efficiency. Intuitively, if the sparsity is 0% or 100%, no new information is introduced during finetuning on the corresponding new/downstream language/task.

**Optimization.** To differentiably optimize $m_t$ in Eq. (1), we binarize $m_t$ and activate only its top $k_t$ elements during forward, while all the elements in $m_t$ are updated via straight-through estimation [100] during backward. More specifically, during forward, we binarize $m_t$ to $\hat{m}_t$ via enforcing its top $k_t$ elements to 1 and other elements to 0, thus the forward function becomes $f(\hat{m}_t \odot \theta_{SSL}, x_t)$, which is the same for the inference process; during backward, we directly propagate the gradients from the binary mask $\hat{m}$ to $m$, i.e., $\frac{\partial l}{\partial m_t} \approx \frac{\partial l}{\partial \hat{m}_t}$, thus $m_t$ can be learned in a gradient-based manner.

### 3.3 How to Initialize the Masks in S³-Router?

**Importance of mask initialization.** A low learning rate is commonly adopted during finetuning to ensure effective inheritance of the SSL speech representations, resulting in merely smaller changes in each set of the masks $m_t$ in Eq. (1) as compared to training from scratch. Therefore, the mask initialization strategy plays an important role for the quality of the finally optimized masks in S³-Router. Next, we discuss the pros and cons of two intuitive mask initialization strategies:

① **Random initialization (RI).** We empirically find that adopting commonly used random initialization [101] in S³-Router can achieve a decent accuracy under a low sparsity. However, since no prior knowledge of the speech SSL model is utilized, the accuracy can largely drop with a high sparsity under random mask initialization, limiting extending S³-Router to be a practical pruning technique.

② **Weight magnitude based initialization (WMI).** As weights magnitude can quantify the importance of weights as commonly used in pruning [102, 103, 104, 105, 106], taking the magnitudes of the given SSL pretrained weights $||\theta_{SSL}||$ in Eq. 1 as the initial values of masks might help utilize

the knowledge learned during SSL pretraining. However, we empirically find that doing so causes worse trainability, i.e., the ranking of mask values is then more stable during finetuning than that under random initialization, and learned binary masks seldom change and mostly stick to their initial values. This may inhibit the optimization process and lead to sub-optimal learned masks.

③ **Proposed Order-Preserving Random Initialization (ORI).** To marry the best of both above initialization strategies, we propose a new one, which can be viewed as a weight rank based initialization. In ORI, we first acquire mask values of the same dimension as the shared weights via random initialization like [101], and then perform magnitude-based sorting to assign a larger mask value to weight elements with a larger magnitude. Thus, the ranking order between the mask values of the weight elements is the same as that of their magnitudes. In this way, the mask trainability is maintained while the learned speech SSL model knowledge can be exploited (see Sec 4).

### 3.4 $S^3$-Router is Useful in Various Application Scenarios

**A new finetuning scheme.** $S^3$-Router offers a new and equally effective finetuning scheme as it finetunes model connections given a speech SSL model. As large-scale speech SSL models tend to overfit when finetuning their weights under a low-resource setting, we hypothesize that the binary optimization of the mask patterns in Eq. (1) can serve as regularization and thus alleviate overfitting.

**An efficient multilingual and multitask method.** $S^3$-Router can naturally enable multilingual and multitask speech processing by merely switching among its learned language-/task-specific binary masks. One advantage is that since the gradients of different spoken languages or tasks can now be independently accumulated on their corresponding masks *without* interfering each other, the commonly observed gradient conflict issue [107] can be alleviated.

**A new pruning technique.** Since $S^3$-Router encodes language/task-specific information via binary masks on top of shared model weights, it naturally introduces sparsity into the given model, and thus can naturally serve as a pruning technique. To further improve the achievable accuracy-efficiency trade-off and more fairly benchmark with SOTA ASR pruning methods, we propose a variant of $S^3$-Router dubbed $S^3$-Router-P, which first finetunes the model weights on the downstream audios and then prunes the model connections based on the learned binary masks in Eq. (1).

**An effective and simple tool to analyze what is encoded across language/task-specific speech SSL models.** Another exciting advantage of $S^3$-Router is that it can provide a quantitative metric about how the pretained SSL speech presentation is utilized by different spoken languages/tasks via masking out languages/tasks-specific weights.

## 4 $S^3$-Router: Discarding $\leq$10% Weights is All You Need

### 4.1 Experiment Setup

Here we evaluate $S^3$-Router as a new finetuning scheme for downstream speech processing.

**Models.** We adopt wav2vec 2.0 base/large (wav2vec2-base/large) [6] and data2vec [108] pretrained on LibriSpeech 960 hours [109] and xlsr [7] pretrained on 128 languages sampled from Common-Voice [110] as our speech SSL models.

**Datasets.** We consider 10 speech processing tasks, including low-resource English ASR on LibriSpeech [7] with only 10min/1h/10h labeled data, following the dataset split in [6, 94], and low-resource phoneme recognition on CommonVoice [110] with 1h labeled data per language, following the dataset split in [111, 7], as well as 8 speech processing tasks from SUPERB [112].

**Finetuning settings.** Our code is built on top of fairseq [113] and we follow the standard finetuning settings for each task, i.e., the default configurations in fairseq [113] for ASR/phoneme recognition and those in SUPERB [112] for other tasks. In particular, all our experiments on ASR/phoneme recognition adopt an Adam optimizer with an initial learning rate of 5e-5 plus a tri-stage schedule [6] and we finetune wav2vec2-base/large for 12k/15k/20k steps on the 10m/1h/10h splits, respectively, and xlsr is finetuned for 12k steps for each spoken language. It takes about 10/24/24 GPU hours to finetune wav2vec2-base/large/xlsr for 12k steps with our $S^3$-Router. We do not freeze all the layers except the final linear layer for the first 10k steps [6], following [94].

**$S^3$-Router settings.** If not specifically stated, $S^3$-Router only tunes the connections of (i.e., applies learnable masks on) the feed-forward networks (FFNs) of transformer structures [114] and fixes all

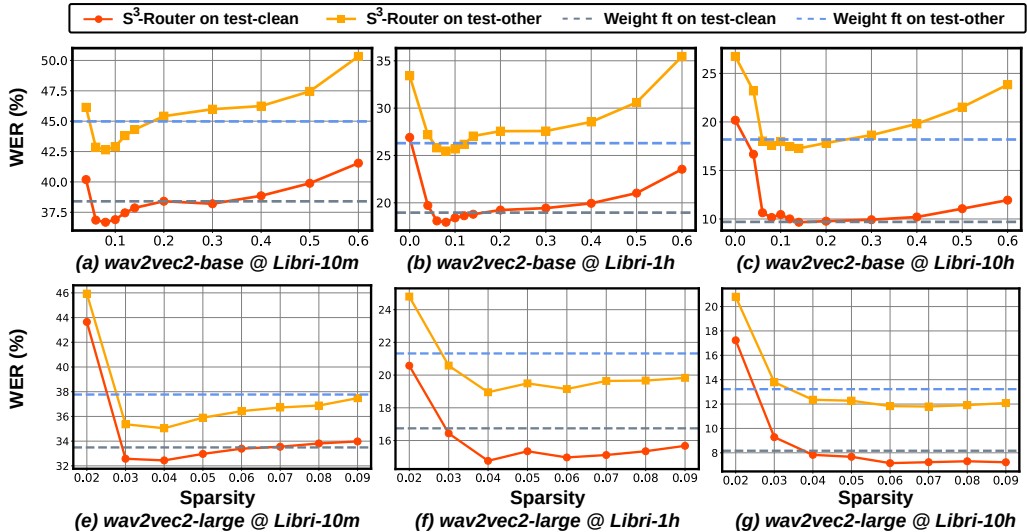

Figure 2: Benchmark our $S^3$-Router and standard weight finetuning on the test-clean/test-other sets of LibriSpeech on top of wav2vec2-base/large under different low-resource settings.

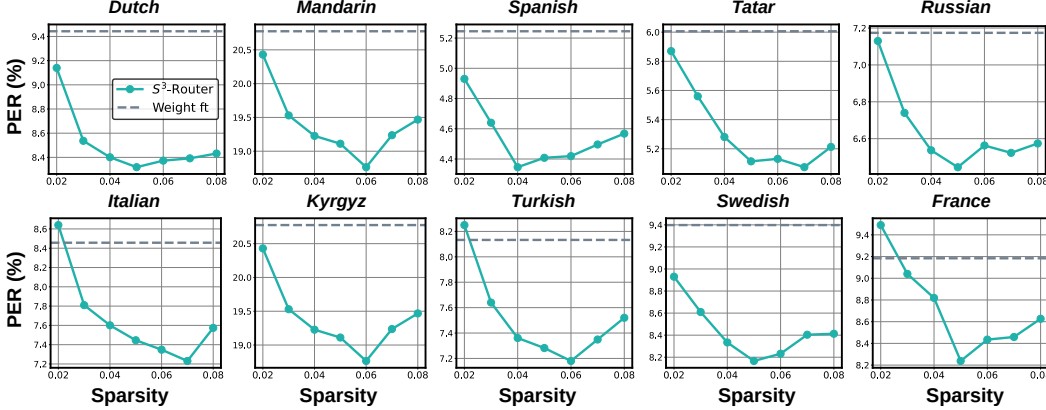

Figure 3: Benchmark our $S^3$-Router and weight finetuning on xlsr across 10 spoken languages.

other weights, which is empirically found to be optimal based on the ablation studies in Sec. 6. In addition, if not stated, we adopt our ORI mask initialization by default and do not apply language models for a fair benchmark of ASR performance, following [94].

## 4.2 Benchmark on Low-resource English ASR

**Finetuning on wav2vec2-base.** We apply our $S^3$-Router on wav2vec2-base under different low-resources settings as shown in Fig. 2 (a)~(c). We can observe that our $S^3$-Router can consistently outperform the standard weight finetuning in terms of the achievable WER, i.e., the lowest WER at the corresponding optimal sparsity. In particular, our $S^3$-Router achieves a 1.72%/2.34% reduction in word error rate (WER) under a sparsity ratio of 8% on the test-clean/test-other set of LibriSpeech, respectively, when being finetuned on 10min labeled data. This indicates that finetuning connections via our $S^3$-Router can be a competitive alternative with reduced WER for finetuning weights under low-resource settings, which provides a new paradigm for finetuning speech SSL models.

**Ablation studies of mask initialization.** We equip our $S^3$-Router with different mask initialization schemes in Sec. 3.3 and show their achievable WER on low-resource English ASR in Tab. 1. We can observe that (1) the proposed ORI mask initialization consistently outperforms the other two schemes in terms of the achievable WER, and (2) even random mask initialization can match or surpass the performance of standard weight finetuning. We also provide their complete sparsity-WER trade-offs in the appendix and find that weight magnitude based initialization favors larger sparsity ratios, while it is harder to overturn the ranking between masks via gradients, resulting in inferior achievable WER.

**Scalability to larger speech SSL models.** We apply $S^3$-Router to a larger model wav2vec-large. As shown in Fig. 2 (e)~(g), we can see that the achievable WER of $S^3$-Router still consistently outperforms the standard weight finetuning across all downstream datasets, e.g., a 1.99%/2.37% WER reduction under a 4% sparsity on the test-clean/test-other set, respectively,

Table 1: Benchmark different mask initialization schemes of $S^3$-Router with weight finetuning on LibriSpeech test-clean/test-other sets.

| Method | Libri-10m | Libri-1h | Libri-10h |
|---|---|---|---|
| Weight ft | 38.40/44.98 | 18.963/26.298 | 9.7/18.19 |
| RI | 36.98/43.42 | 18.19/26.31 | 9.78/18.17 |
| WMI | 40.09/46.63 | 19.99/27.16 | 11.17/18.97 |
| **ORI (Ours)** | **36.68/42.64** | **17.95/25.47** | **9.66/17.27** |

when being finetuned on the 1h labeled data. Consistent results on xlsr are provided in the appendix.

Insights. This set of experiments indicates that ❶ our $S^3$-Router features a good scalability to larger speech SSL models as well as a good generality for different pretraining schemes, and ❷ our method effectively reduces the overfitting on more overparameterized speech SSL models according to the larger performance gains, thanks to the regularization effect of the binary optimization process in Eq. (1), making our method an appealing solution for more advanced speech SSL models.

**Add language models (LMs).** We further apply a 4-gram LM [115] as the decoder for both our $S^3$-Router at the optimal sparsity ratio in Fig. 2 and the weight finetuning baselines. In particular, we adopt the official 4-gram language model (LM) [115] with a beam size of 50, an LM weight of 2, and a word insertion penalty of -1. As shown in Tab. 2, our $S^3$-Router still

Table 2: Benchmark our $S^3$-Router with standard weight finetuning when being equipped with a 4-gram LM [115].

| Model | Method | Libri-10m | Libri-1h | Libri-10h |
|---|---|---|---|---|
| wav2vec2 -base | Weight ft | 26.55/32.95 | 11.54/18.45 | 6.65/13.97 |
| | $S^3$-**Router** | **25.28/31.07** | **11.28/18.33** | **6.42/13.14** |
| wav2vec2 -large | Weight ft | 24.83/29.05 | 11.18/15.80 | 5.52/10.31 |
| | $S^3$-**Router** | **23.17/25.83** | **9.74/13.77** | **4.98/9.27** |

consistently outperforms weight finetuning with more notable reductions on wav2vec2-large, e.g., a 1.66%/3.22% WER reduction on the test-clean/test-other set of LibriSpeech when being finetuned on 10min labeled data.

**Scalability to speech models pretrained by other SSL paradigms.** To validate the generalization capability of our $S^3$-Router across speech models pretrained by other SSL paradigms, we further apply our method on the SOTA speech SSL model data2vec [108] under different sparsity ratios, which features a new SSL pretraining paradigm based on self-distillation. Here we follow the same finetuning setting as wav2vec2-base, which is also the de-

Table 3: Benchmark our $S^3$-Router under different sparsity ratios with weight finetuning on top of data2vec on LibriSpeech test-clean/test-other sets.

| Method | Libri-10m | Libri-1h | Libri-10h |
|---|---|---|---|
| Standard ft | 30.75/34.612 | 14.15/19.61 | 7.28/13.11 |
| $S^3$-Router@0.07 | 30.78/35.17 | 14.09/19.72 | 7.56/13.39 |
| $S^3$-Router@0.08 | 30.70/34.45 | 13.96/19.41 | 7.43/13.34 |
| $S^3$-Router@0.09 | **29.86/34.09** | **13.92/19.43** | 7.23/13.25 |
| $S^3$-Router@0.10 | 31.30/35.07 | 14.27/20.10 | **7.05/12.98** |

fault finetuning setting of data2vec in fairseq [113]. As shown in Tab. 3, we can observe that our $S^3$-Router still achieves lower WER across all resource settings, e.g., a 0.89% WER reduction on LibriSpeech test-clean when being trained with 10m labeled speech. This indicates that our $S^3$-Router can generally serve as a competitive finetuning paradigm independent of the SSL scheme.

### 4.3 Benchmark on Low-resource Cross-lingual Transfer

We further evaluate our $S^3$-Router under two cross-lingual transfer settings, including (1) finetuning the multilingual pretrained xlsr on different low-resource languages in CommonVoice [110], and (2) a high-to-low resource transfer setting where the monolingual (English) pretrained wav2vec2-base is finetuned on multiple low-resource languages in CommonVoice.

**Cross-lingual transfer on xlsr.** As shown in Fig. 3, our $S^3$-Router consistently outperforms standard weight finetuning in terms of the achievable phoneme error rate (PER) across all the 10 spoken languages, e.g., a 1.12%/2.01% PER reduction on Dutch/Mandarin, respectively.

**High-to-low resource transfer on wav2vec2-base.** As shown in Tab. 4, on top of the English pretrained wav2vec2-base, finetuning the connections with $S^3$-Router still wins the lowest achievable PER over the baseline. Consistent results on wav2vec2-large are in the appendix.

Insights. This set of experiments implies that for each downstream spoken language, there exist

Table 4: Benchmark our $S^3$-Router and weight finetuning on wav2vec2-base and CommonVoice.

| Language | Dutch | Mandarin | Spanish | Tatar | Russian |
|---|---|---|---|---|---|
| Weight ft | 19.82 | 26.67 | 13.86 | 11.14 | 17.05 |
| $S^3$-Router | **18.51** | **26.10** | **13.37** | **10.94** | **16.33** |

| Language | Italian | Kyrgyz | Turkish | Swedish | France |
|---|---|---|---|---|---|
| Weight ft | 19.27 | 13.41 | 15.70 | 20.81 | 19.35 |
| $S^3$-Router | **18.29** | **12.30** | **14.82** | **19.64** | **17.94** |

Table 5: Benchmark S$^3$-Router with standard weight finetuning on 8 tasks from SUPERB [112].

| Catagory | Content | Speaker | | | Paralinguistics | Semantics | | |
|---|---|---|---|---|---|---|---|---|
| Task | Keyword Spotting | Speaker Identification | Speaker Verification | Speaker Diarization | Emotion Recognition | Intent Classification | Slot Filling | Speech Translation |
| Metric | Acc ↑ | Acc ↑ | EER ↓ | DER ↓ | Acc ↑ | Acc ↑ | F1 ↑ | BLEU ↑ |
| Weight ft | 95.5 | 66.8 | 7.24 | 7.41 | 61.5 | 91.48 | 88.1 | 18.85 |
| S$^3$-Router | **95.7** | **71.07** | **6.79** | **7.18** | **62.13** | **92.6** | **88.76** | **19.01** |
| Opt Spar. | 0.1 | 0.06 | 0.1 | 0.08 | 0.1 | 0.1 | 0.1 | 0.08 |

decent subnetworks for processing its language-specific information even in monolingual pretrained speech SSL models. In another words, properly learned sparsity can encode language-specific information with competitive ASR performance.

### 4.4 Benchmark on More Downstream Speech Processing Tasks

We further evaluate our S$^3$-Router via finetuning wav2vec2-base on 8 speech processing tasks from SUPERB [112], covering different aspects of speech (content/speaker/semantics/paralinguistics). We show the achievable task performance as well as the corresponding optimal sparsity ratios in Tab. 5 and find that our method surpasses the task performance over standard weight finetuning across all 8 tasks via discarding ≤10% weights, indicating that properly learned sparsity can also effectively encode task-specific information. We provide the sparsity-performance trade-offs in the appendix.

### 4.5 Empowering Multilingual and Multitask Speech Processing

Since all the aforementioned results achieved on the same speech SSL model via S$^3$-Router share pretrained weights, it can simultaneously enable multilingual and multitask speech processing, thanks to the independent accumulation of task-specific gradients that avoids gradient conflicts [107], of superior scalability with the number of tasks. For example, as compared to independent weight finetuning for each language/task, S$^3$-Router can simultaneously support 11 languages in Sec. 4.2/ 4.3 and 8 tasks in Sec. 4.4 using one wav2vec2-base while achieving a win-win in both accuracy (as validated in Sec. 4.2/ 4.3/ 4.4) and efficiency, i.e., more than 88.5% reductions in model parameters.

### 4.6 Benchmark S$^3$-Router with Adaptor Tuning

We further benchmark our S$^3$-Router with adaptor tuning [55], which has emerged as an alternative finetuning scheme. In particular, we reproduce the speech adaptor design in [116], following their open-sourced implementation, on wav2vec2-base and benchmark with our reported results of S$^3$-Router under the best sparsity settings for different languages. As shown in Tab. 6, we can observe that our S$^3$-Router still consistently achieves the lowest WER/PER across all languages and resource settings, e.g., a 5.14%/3.24% WER reduction on LibriSpeech test-clean when being trained on LibriSpeech-10m/1h, respectively.

Table 6: Benchmark our S$^3$-Router with adaptor tuning [116] on top of wav2vec2-base across LibriSpeech test-clean/other and CommonVoice.

| Method | Libri-10m | Libri-1h | Libri-10h |
|---|---|---|---|
| Standard ft | 38.40/44.98 | 18.96/26.30 | 9.70/18.19 |
| Adaptor [116] | 41.82/49.01 | 21.19/28.92 | 12.26/19.67 |
| S$^3$-Router | **36.68/42.64** | **17.95/25.47** | **9.77/17.27** |

| Method | Dutch | Spanish | Mandarin |
|---|---|---|---|
| Standard ft | 19.82 | 13.86 | 26.67 |
| Adaptor [116] | 22.63 | 15.89 | 29.03 |
| S$^3$-Router | **18.51** | **13.37** | **26.10** |

Insight. Under a low-resource setting, it is hard for adaptor tuning to maintain a comparable accuracy over standard weight finetuning, while our S$^3$-Router often outperforms standard weight finetuning as shown in Sec. 4.2, thanks to its binary optimization process on the masks, which can potentially regularize the learning process and lead to better downstream performances as compared to finetuning the overparameterized model weights. This indicates that S$^3$-Router is a better finetuning scheme over adaptor tuning, especially under low-resource settings.

## 5 S$^3$-Router-P: Pruning ASR Models for Enhancing Efficiency

**Setup.** For evaluating S$^3$-Router-P (see Sec. 3.4), we conduct two modifications: (1) both FFN and self-attention (SA) modules are pruned for a fair comparison, and (2) we adopt weight magnitude

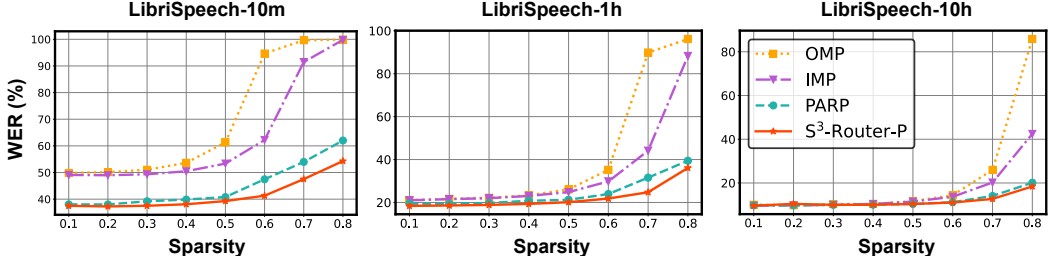

Figure 4: Benchmark our $S^3$-Router-P against OMP, IMP, and PARP [94] for pruning wav2vec2-base on LibriSpeech. The WER on the test-clean set is reported.

based mask initialization for two reasons: firstly, it features the best scalability to high sparsity among the three initialization schemes; secondly, since the weights have been finetuned, their magnitudes can serve to indicate their importance on the downstream speech. Ablation studies for pruning under different mask initialization schemes are in the appendix.

For our baselines, we benchmark with three ASR pruning methods, i.e., one-shot/iterative magnitude pruning (OMP/IMP), and the SOTA method PARP [94] under the best setting (dubbed PARP-P), and directly adopt their reported results in [94].

**Pruning wav2vec2-base on LibriSpeech.** As shown in Fig. 4, we can observe that (1) our $S^3$-Router-P consistently achieves the most competitive WER-sparsity trade-offs across the two models and three datasets, e.g., a 6.46% lower WER over PARP under a sparsity ratio of 0.7 with 10min labeled data, and (2) our method features a better scalability to more stringent low-resource scenarios according to the large performance gains on LibriSpeech-10m.

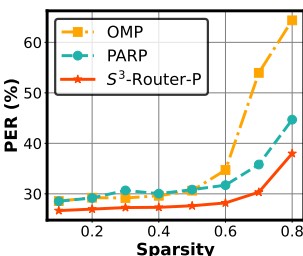

Figure 5: Benchmark our $S^3$-Router-P against OMP and PARP for pruning on Mandarin.

**Pruning wav2vec2-base on Mandarin@CommonVoice.** As shown in Fig. 5, consistent observations can be drawn that our method still wins the WER-sparsity trade-offs. More pruning results are provided in the appendix.

## 6  $S^3$-Router: What is Encoded in Speech SSL Models?

### 6.1  What is The Roles of Different Modules in Speech SSL Models?

We tune the connections of a subset of the modules in the speech SSL model via our $S^3$-Router to study their contributions with other modules fixed in terms of both weights and connections.

**FFN vs. SA.** We tune the connections of FFN, SA, or both on top of wav2vec2-base and LibriSpeech-1h as shown in Tab. 7. We can observe that tuning the connections of FFN only wins the lowest achievable WER, even outperforming tuning both FFN and SA. This indicates that the SSL pretrained SA modules are generalizable enough to capture temporal relationship between tokens for downstream speech thus only FFN needs to be tuned for encoding new information about downstream tasks.

Table 7: The achievable WER on LibriSpeech test-clean/test-other when finetuning different modules.

| Sparsity | SA only | FFN only | Both |
|---|---|---|---|
| 0.1 | 24.39/30.79 | **18.62/26.12** | 19.26/27.36 |
| 0.2 | 23.06/30.22 | **19.54/27.07** | 18.99/27.38 |
| 0.3 | 22.78/30.21 | **19.59/27.58** | 19.65/28.40 |
| 0.4 | 23.22/31.28 | 20.02/28.97 | **19.78/27.98** |

**Roles of different blocks.** To study the roles of the blocks at different depths, we uniformly divide the 12 blocks in wav2vec2-base into 4 groups and only tune the connections of one group. As shown in Tab. 8, we indicate the tunable group as 1 and other fixed ones as 0. We can observe that generally tuning later groups achieves lower WER and tuning the 1st/2nd groups only will result in unacceptably high WER, aligning with our intuition that early blocks in speech SSL models extract generalizable phonetic features while later blocks capture task-specific features thus are required to be tuned. In

Table 8: Apply $S^3$-Router on different groups of blocks. WERs on LibriSpeech test-clean/test-other are reported.

| Group | Libri-1h | Libri-10h |
|---|---|---|
| [1,0,0,0] | 85.68/92.12 | 77.65/86.12 |
| [0,1,0,0] | 54.96/67.17 | 42.26/56.24 |
| [0,0,1,0] | **24.57/32.68** | **14.95/22.96** |
| [0,0,0,1] | 35.12/42.39 | 20.88/27.29 |

addition, tuning the 3rd group achieves the lowest WER, even close to that of tuning all groups, which we assume is because the features starting from the 3rd group are task-specific and not generalizable across downstream tasks thus demandingly require to be finetuned.

## 6.2 How Much New Information is Learned by Finetuning?

We visualize the optimal sparsity ratio for achieving the lowest WER on LibriSpeech-10m/1h/10h, which serves as an indicator about the amount of new information learned by finetuning, across wav2vec2-base/large/xlsr in Tab. 9. We can observe that (1) larger speech SSL models reach their optimal performance under lower sparsity, i.e., relatively less amount of tuning could lead to their decent downstream performance thanks to their overparameterization, and (2) finetuning on more resources leads to higher optimal sparsity, indicating that speech SSL models have more confidence to update the SSL pretrained representations given more resources.

Table 9: The optimal sparsity ratios for the lowest WER across different models and resources.

| Model | Libri-10m | Libri-1h | Libri-10h |
|---|---|---|---|
| wav2vec2-base | 0.08 | 0.10 | 0.20 |
| wav2vec2-large | 0.04 | 0.04 | 0.06 |
| xlsr | 0.04 | 0.05 | 0.07 |

## 6.3 How is The Learned Masks Correlated to Phonetics?

Given the decent performance achieved by the learned masks, we are interested their correlations with human expertise in phonetics. In particular, we wonder whether the similarity between the learned masks of two languages aligns with that between their phoneme inventories.

**Setup.** Given the 11 languages adopted in Sec. 4.2 and 4.3, we calculate layer-wise cosine similarities of their learned masks, under a sparsity ratio of 0.1 with near-optimal performance, between each pair of them. We pick the mask similarity of the first layer as our metric without losing generality. We further measure the cosine similarity between their corresponding phoneme inventories acquired from Phoible [117], a cross-linguistic phonological inventory database which covers over 2000 languages. Following [118], we combine the inventories for all languages to a shared phoneme inventory and use a binary vector to indicate the phone inventory of each language, thus the language-wise phonetic similarities can be calculated as the cosine similarities between their corresponding binary vectors.

**Results and analysis.** We visualize the two similarity metrics of each language pair in Fig. 6 and find that the Pearson Correlation Coefficient [119] and the Spearman correlation coefficient [120] between the two similarity metrics are 0.527 and 0.548, respectively. This indicates that the similarities of our learned masks are nontrivially correlated with that of phonetics while the former also provides new insights in the eyes of speech SSL models, which could shed light on future advances in zero-shot cross-lingual transfer [118]. We provide more insightful correlation analysis as well as the visualization of layer-wise mask similarities between different languages in the appendix.

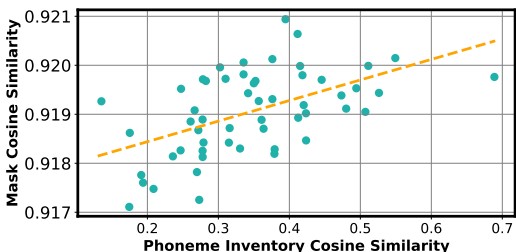

Figure 6: Visualizing the correlation between the similarity of the learned masks and that of the corresponding phoneme inventories of language pairs.

## 7 Conclusion

Motivating by the empirical success of SSL speech representations in low-resource speech processing, we propose $S^3$-Router to facilitate the practical usage of speech SSL models via finetuning their connections instead of weights and encoding language-/task-specific information via sparsity. Extensive experiments validate that $S^3$-Router not only serves as a stronger alternative with alleviated overfitting and enhanced accuracy for standard weight finetuning, but also empowers efficient multilingual and multitask speech processing. Our insights could enhance our understandings about what is encoded in speech SSL models and thus shed light on future advances in SSL speech representations.

## Acknowledgements

The work is supported by the National Science Foundation (NSF) through the CCRI program (Award number: 2016727) and an IBM faculty award received by Dr. Yingyan Lin.

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
