# Supplementary Materials of
# Losses Can Be Blessings: Routing Self-Supervised Speech Representations Towards Efficient Multilingual and Multitask Speech Processing

**Yonggan Fu[1], Yang Zhang[2], Kaizhi Qian[2], Zhifan Ye[3], Zhongzhi Yu[1]**
**Cheng-I Lai[4], Yingyan (Celine) Lin[1]**
[1]Georgia Institute of Technology, [2]MIT-IBM Watson AI Lab, [3]Rice University, [4]MIT CSAIL
{yfu314,zyu401,celine.lin}@gatech.edu
{yang.zhang2,kqian}@ibm.com {zy50}@rice.edu {clai24}@mit.edu

## 1 Overview and Outline

In this supplement, we provide more experiments and analysis as a complement to the main content, which are outlined below:

- We provide more results of our $S^3$-Router as a new finetuning paradigm in Sec. 2, including xlsr on LibriSpeech and wav2vec2-large on CommonVoice, and the sparsity-WER trade-offs achieved by different mask initialization schemes or on different speech processing tasks;

- We provide more results of our $S^3$-Router for ASR pruning in Sec. 3, including an ablation study of different mask initialization schemes and more benchmarks with PARP [1] for ASR pruning on more speech SSL models;

- We study more properties of speech SSL models via our $S^3$-Router in Sec. 4, including the layer-wise similarity of learned masks between different languages and more analysis regarding the correlation between the learned masks of and phonetics;

## 2 More Evaluation Results of $S^3$-Router as a New Finetuning Paradigm

**Finetuning xlsr on LibriSpeech.** We benchmark our $S^3$-Router with standard weight finetuning on top of the multilingual SSL pretrained xlsr for low-resource English ASR on LibriSpeech. As shown in Fig. 1, our $S^3$-Router outperforms standard weight finetuning in terms of the achievable WER, e.g., a 6.02%/8.33% reduction in WER on LibriSpeech test-clean/other when being finetuned on 10min labeled data, respectively, indicating the decent scalability and generality of our $S^3$-Router even under the large gap between the pretraining resources and finetuning resources.

**Finetuning wav2vec2-large on CommonVoice.** We benchmark our $S^3$-Router with standard weight finetuning on top of English pretrained wav2vec2-large for phoneme recognition on other languages from CommonVoice, which is a high-to-low resource transfer setting as a complement to Sec. 4.3 of our manuscript. As shown in Tab. 1, consistent with the results in our manuscript, our $S^3$-Router still wins the lowest achievable PER over weight finetuning, which further validates the scalability of our method to larger models under cross-lingual transfer settings.

Table 1: Benchmark our $S^3$-Router and weight finetuning on wav2vec2-large and CommonVoice.

| Language | Dutch | Mandarin | Spanish | Tatar | Russian |
|---|---|---|---|---|---|
| Weight ft | 19.821 | 26.674 | 13.864 | 11.143 | 17.052 |
| $S^3$-Router | **17.327** | **25.897** | **11.887** | **9.912** | **15.403** |
| Language | Italian | Kyrgyz | Turkish | Swedish | France |
| Weight ft | 19.265 | 13.409 | 15.699 | 20.807 | 19.349 |
| $S^3$-Router | **17.425** | **11.279** | **13.31** | **18.664** | **16.304** |

36th Conference on Neural Information Processing Systems (NeurIPS 2022).

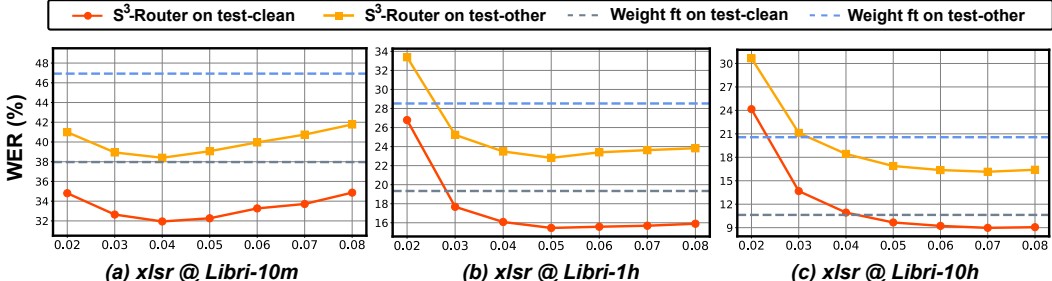

Figure 1: Benchmark our S³-Router and standard weight finetuning on the test-clean/test-other sets of LibriSpeech on top of the multilingual pretrained xlsr under different low-resource settings.

**Complete sparsity-WER trade-offs of different mask initialization schemes.** We provide the complete sparsity-WER trade-offs achieved different mask initialization schemes for finetuning wav2vec2-based on LibriSpeech-1h with our S³-Router, as a complement to Tab.1 in Sec. 4.2 of our manuscript, in Fig. 2, In addition to the lowest achievable WER of our proposed ORI mask initialization, we can also observe that (1) the achievable WER of random mask initialization can match or surpass that of standard weight finetuning under small sparsity ratios while it suffers from a steep increase in WER along with the increase in sparsity ratios due to the lack of utilizing the priors of the pretrained speech representation; (2) Weight magnitude based initialization features better scalability to large sparsity ratios, where keeping more important connections, approximately indicated by higher weight magnitudes, becomes more crucial. However, its achievable WER is even inferior to that of random mask initialization due to worse trainability, i.e., it is harder to overturn the ranking between masks via gradients, especially when magnitudes of the pretrained weights cannot serve as a good metric for indicating their importance on downstream speech. Our proposed ORI strategy marries the best of both worlds thus wins the best achievable WER.

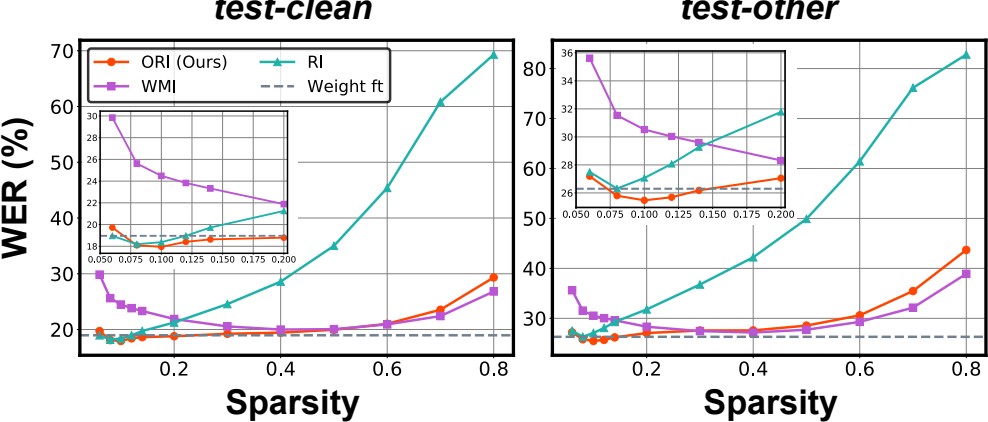

Figure 2: Benchmark different mask initialization schemes for finetuning wav2vec2-base on top of LibriSpeech-1h via our S³-Router. Both WERs on test-clean (left) and test-other (right) are reported.

**Complete sparsity-WER trade-offs on different speech processing tasks.** We also provide the complete sparsity-WER trade-offs for finetuning wav2vec2-base via our S³-Router on different speech processing tasks from SUPERB [2] in Tab. 5/Sec. 4.4 of our manuscript. In particular, we pick one representative task from each of the four task categories for processing different aspects of speech (content/speaker/semantics/paralinguistics). As shown in Fig. 3, we can make a consistent observation as Sec.4 of our manuscript that properly discarding $\leq 10\%$ weights can serve as a decent alternative, featuring better achievable task performances, for weight finetuning.

## 3  More Evaluation Results of S³-Router as an ASR Pruning Technique

**Pruning xlsr/wav2vec2-base on CommonVoice.** We further evaluate our S³-Router-P for ASR pruning on more speech SSL models as a complement to Sec. 5 of our manuscript. As shown in Fig. 4, our S³-Router-P consistently achieves comparable or better pruning performances over PAPR

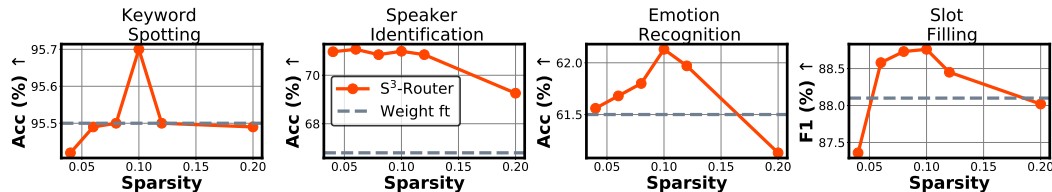

Figure 3: Benchmark our $S^3$-Router with standard weight finetuning on top of wav2vec2-base on four representative speech processing tasks from SUPERB [2].

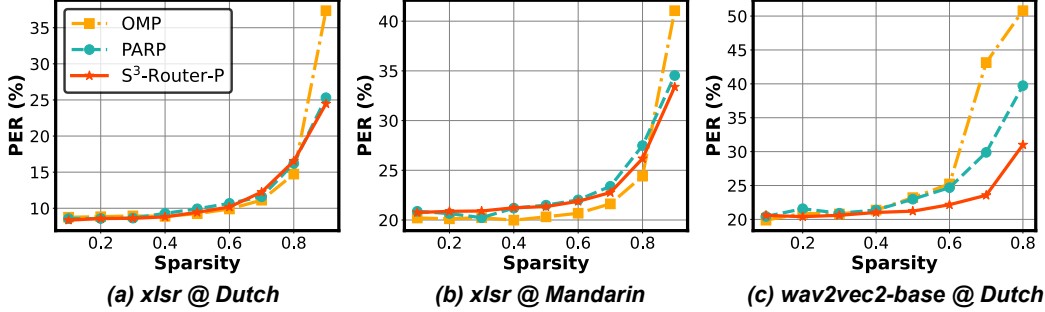

Figure 4: Benchmark our $S^3$-Router-P against OMP and PARP for pruning xlsr/wav2vec2-base on Dutch/Mandarin from CommonVoice.

under large sparsity ratios across different models and downstream datasets, which indicates the potential of our method as an all-in-one technique for facilitating the practical usage of speech SSL models.

**Ablation study of the impact of different mask initialization schemes for ASR pruning.** We adopt different mask initialization schemes in our $S^3$-Router for ASR pruning and benchmark their pruning performances in Fig. 5. We can observe that weight magnitude based mask initialization wins better scalability to large sparsity ratios over ORI, although the latter achieves better finetuning performances as shown in Sec. 2. We assume this is because after weight finetuning on downstream speech, the magnitudes of model weights can serve as a better indicator for their importance on the downstream speech, resulting in the better scalability to large sparsity ratios where keeping more important connections becomes more crucial. Therefore, we adopt weight magnitude based mask initialization for ASR pruning by default in our manuscript.

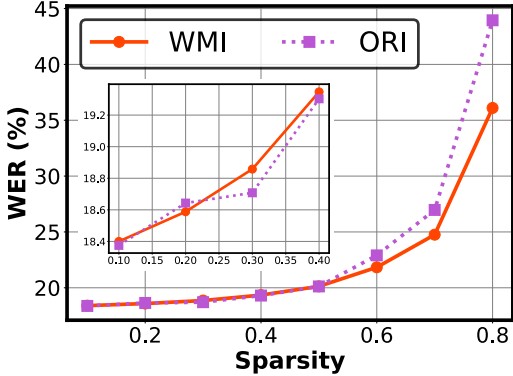

Figure 5: Benchmark different mask initialization schemes for pruning wav2vec2-based on top of LibriSpeech-1h. WERs on test-clean are reported.

## 4 More Analysis about the Properties of Speech SSL Models via $S^3$-Router

**Layer-wise cosine similarity of learned masks between different languages.** We visualize the layer-wise cosine similarity between the learned mask, under a sparsity ratio of 0.1 with near-optimal performances, for English ASR on LibriSpeech-1h and the ones learned on six languages from CommonVoice in Fig. 6. In particular, there are totally 24 tunable fully-connected layers in the 12 FFNs of wav2vec2-base. We can observe that (1) the learned masks are similar across languages with cosine similarities $\geq 0.9$; (2) Later blocks generally feature lower mask similarities, indicating that more language-specific information are encoded in later blocks; (3) The mask similarity ranking keeps relatively stable across consecutive layers; (4) The mask similarity generally aligns with human intuitions, e.g., the mask similarity between English and Dutch is higher than that between English and Mandarin, and the quantitative degree of such alignment is indicated by the correlation between their mask similarity and their expert-defined similarity in phonetics.

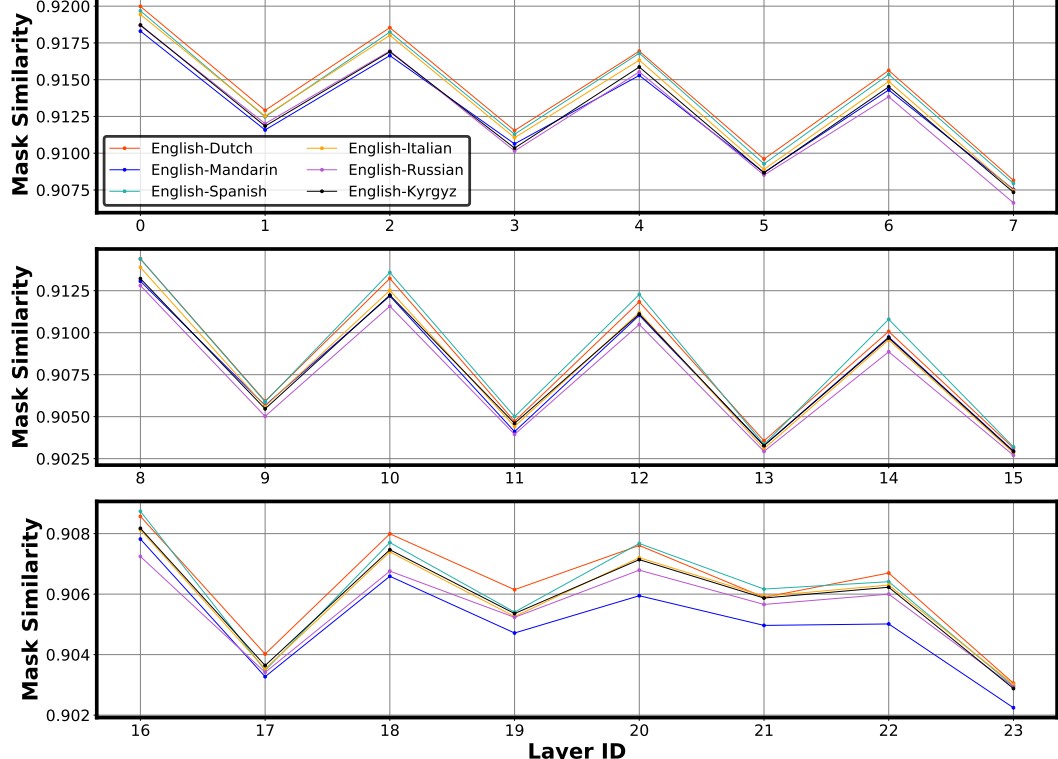

Figure 6: Visualizing the layer-wise cosine similarity of learned masks, on top of the 24 layers in 12 FFNs of wav2vec2-base, between different languages.

**Correlation between mask similarities and phonetic similarities calculated on different layers.** As shown in Sec. 6.3 of our manuscript, there exists a non-trivial correlation between the two similarity metrics on the first layer. We further measure the Pearson Correlation Coefficient [3] between the phonetic similarity and the mask similarity calculated on top of different layers. As shown in Tab. 2, we can observe non-trivial correlations between the two similarity metrics on early layers while their correlations on later layers are poor. This further indicates that early layers in speech SSL models process phonetic features extracted from raw audios, which are highly correlated to human expertise in phonetics, while later blocks process more task-specific features, which integrate other high-level information, e.g., for language modeling in ASR.

Table 2: Pearson correlation between two similarity metrics calculated on different layers.

| Layer ID | 0 | 1 | 2 | 3 | 4 | 5 |
|---|---|---|---|---|---|---|
| Pearson Corr | 0.527 | 0.374 | 0.585 | 0.477 | 0.611 | 0.431 |
| **Layer ID** | 6 | 7 | 8 | 9 | 10 | 11 |
| Pearson Corr | 0.561 | 0.372 | 0.539 | 0.366 | 0.436 | 0.351 |
| **Layer ID** | 12 | 13 | 14 | 15 | 16 | 17 |
| Pearson Corr | 0.416 | 0.271 | 0.296 | 0.17 | 0.125 | 0.126 |
| **Layer ID** | 18 | 19 | 20 | 21 | 22 | 23 |
| Pearson Corr | 0.156 | 0.105 | 0.225 | 0.211 | 0.154 | 0.118 |