# OpenReview forum: "Losses Can Be Blessings: Routing Self-Supervised Speech Representations Towards Efficient Multilingual and Multitask Speech Processing"
_NeurIPS.cc/2022/Conference — NeurIPS 2022 Accept_

### Official Review · Reviewer_13fn · 2022-07-11

**Rating:** 5
**Confidence:** 4
**Soundness:** 1 poor
**Presentation:** 3 good
**Contribution:** 2 fair

**Summary:**

This paper proposed a novel method that fine-tunes speech SSL models by discarding the connections rather than changing the weight values. Experiments were conducted on simulated low-resource multilingual ASR tasks as well as several other speech tasks. An interesting finding was that the learnt masks were linearly correlated to the phone inventories of each individual language.

**Questions:**

1. The authors claimed that
"where the speech SSL models suffer from a higher risk of overfitting on downstream low-resource tasks as compared to NLP"
Is there an evidence or proof for this?

2. Similarly,
"As large-scale speech SSL models tend to overfit when finetuning their weights under a low-resource setting, we hypothesize that the binary optimization of the mask patterns in Eq. (1) can serve as regularization and thus alleviate overfitting"
Do the authors think regularisation methods (such as L1/L2 regularisors, dropout, early stopping, and smaller learning rates, or simply only fine-tune a small parts of the model) will make a difference here?



**Ethics Review Area:**

["I don’t know"]

**Limitations:**

1. Compare with other adaptation/adaptor methods.
2. Study other types of speech-based SSL models, if the findings are claims to be generic.
3. Based on the current hardware, routing within sparse models (e.g. MoE models or Pathway models) would be more efficient than routing each individual connection within a densely-connected model, unless the findings indicate way more connections can be discarded (e.g. more than 70%) -- this would allow to use sparse rather than dense matrix structures and ops.

**Strengths And Weaknesses:**

Strength:
1. The paper is very carefully written and easy to follow.
2. The idea of discarding connections is possibly still novel for speech SSL models.
3. Many experiments were done and mostly the results were positive.

Weaknesses:

1. In an alternative view, the proposed S3-Router method is a binary-mask-based task adaptor approach. As the authors claimed, mainly it required to build such masks for all fully-connected layers, which results in a large number of trainable adaptor parameters, in addition to the task-specific output layers. This is a very inefficient way to build such adaptors compared to many other existing alternatives. So the more proper baselines to compare with, are other alternative adaptation methods instead of simply fine-tuning all model encoder parameters.

2. Although it is claimed that

"However, there exists a dilemma between the trends of speech SSL models and the growing demand for speech processing applications on the edge. While advanced speech SSL models become increasingly larger to learn more generalizable features, it is highly desired to process the captured speech signals in real-time on edge devices, which have limited resources and conflict with the prohibitive complexity of existing speech SSL models."

The S3-Router actually increases computational cost rather than reduces it, since masks are required to be applied every time, and the masked weights still need to be used via normal GEMM functions due to the design of the current hardware.
This also results in high storage costs. Users either need to store the resulted many binary mask parameters and apply them every time with extra computation cost (although the binary masks can be stored in bits), or an extra set of parameters need to be stored for every task, which contradicts with the claim

"For example, compared to independent weight fine-tuning for each language/task, S3-Router can simultaneously support 11 languages in Sec. 4.2/ 4.3 and 8 tasks in Sec. 4.4 using one wav2vec2-base while achieving a win-win in both accuracy (as validated in Sec. 4.2/ 4.3/ 4.4) and efficiency, i.e., more than 88.5% reductions in model parameters."

I believe other task adaptation methods can be way more efficient in terms of both computation and storage costs. Most of the advantages the authors claimed are general benefits of using adaptors/adaptations (e.g. "One exciting advantage is that since the gradients of different spoken languages or tasks can..."), which have been studied extensively in the speech community in the past three decades (& these works are completely ignored by this paper).

3. The results are also disappointing. Although so many extra parameters are introduced, it was found that only 10% of the connections can be discarded, which is not "non-trivial contributions (claimed by the authors)" and possibly not useful in practice at all.  In general, I feel this paper overclaims.

4. Although the findings were claimed to be generic (e.g. discarding less than 10% connections), actually only one type of speech SSL model (Wav2Vec 2 related models) was studied, which is not convincible.

5. The results were not compared with other published numbers on the same tasks (e.g. those from SUPERB).

---

> ### Author Response · Authors · 2022-08-02
> **Author Response to Reviewer 13fn (Part 1)**
>
> Thanks for your review comments and we have addressed all your comments and concerns as follows:
>
> **1. Clarification of our method, contributions, and results:**
>
> We first clarify your potential misunderstandings of our method, contributions, and results, and  emphasize the following points:
>
> **(1)** As recognized by both the other two Reviewers as “quite interesting”/“interesting and impactful approach”/“be of interest to the community”, our method is the first to find that task-specific information can be decoded by setting some weights to zero in pretrained speech SSL models, while all pretrained weights are kept fixed and not finetuned. That is why “discarding no more than 10% of the connections” is recognized to be “quite interesting” by Reviewer nV4F and “interesting and impactful approach” by Reviewer 2h6A, as it indicates that SSL pretrained speech models have encoded generalizable features and naturally contained sufficient information required by downstream tasks, e.g., simply zeroing out no more than 10% of the connections can effectively decode task-specific information encoded within SSL pretrained speech models. To the best of our knowledge, we are the first to observe and exploit such properties of speech SSL models, which could enhance the understanding of what is encoded in speech SSL models, as shown in Section 6 of our manuscript.
>
> **(2)** Note that in terms of single-task finetuning, although the binary masks are learned by the binary optimization in Eq.(1) of our manuscript, during inference the weights indicated as zero by the binary mask can be simply set to zero, thus the inference process is the same with the original model (if such sparsity is not utilized by the hardware).
>
> **(3) More importantly, we demonstrate the effectiveness of our method as a pruning technique in Figures 4 and 5 in Section 5 of our submitted manuscript, as well as Section 3 in our supplementary material, and show that our method can achieve better or comparable pruning results as compared to the SOTA pruning method PARP (NeurIPS 2021, according to reference [88]) on speech SSL models.** For example, as shown in Figure 4 of our manuscript, our method can achieve 12.711 WER on LibriSpeech test-clean under a sparsity of 70%, when being finetuned on LibriSpeech-10h, which surpasses all other pruning baselines. This 70% sparsity matches what you mentioned “unless the findings indicate way more connections can be discarded (e.g. more than 70%)” and more results can be obtained in the paper.
>
> Therefore, we understand that your comment of “the results are also disappointing” may be caused by misunderstandings, while the other two reviewers recognize our results as “the experimental results look strong” and “thorough evaluation”.  We will further emphasize our key spirit and contributions in the final version.

---

> > ### Author Response · Authors · 2022-08-02
> > **Author Response to Reviewer 13fn (Part 2)**
> >
> > **2. Comparison with other adaptors for speech:**
> >
> > Thanks for the suggestion! We emphasize that the key motivation for our work is to **(1)** explore and exploit the role of sparsity in speech SSL models for understanding what is encoded within the models and **(2)** improve their inference efficiency, while previous adaptors mostly aim to improve the training efficiency via reducing the numbers of trainable parameters, i.e., the target efficiency metric is different.
> >
> > We agree that adding the comparison with other adaptors can further strengthen our experiment. Following your suggestion, we benchmark our method with two recent speech adaptor designs [1][2] (listed below and [1] is the most recent speech adaptor design) since they explicitly target ASR or have open-source codes among all existing speech adaptors.
> >
> > We first compare S3-Router with the reported results in [1] for wav2vec2-base on LibriSpeech dev-clean/other. Both our method and the baseline are trained on LibriSpeech-10h and we adopt a sparsity ratio of 0.14 for S3-Router according to Figure 2(c) of our manuscript. As shown in the table below, our S3-Router achieves the lowest WER, e.g., a reduction of 0.68% on dev-clean over Adaptor [1].
> >
> > | **Method** | **Standard ft** | **Adaptor [1]** | **S3-Router** |
> > |:---:|:---:|:---:|:---:|
> > | dev-clean | 8.98 | 9.39 | **8.71** |
> > | dev-other | 16.90 | 17.00 | **16.88** |
> >
> > We also reproduce [2], following their open-sourced implementation, on wav2vec2-base and benchmark with our reported results of S3-Router under the best sparsity settings for different tasks shown in Section 4 of our manuscript. As shown in the table below (both results obtained on test-clean/other are reported), we can observe that our S3-Router still consistently achieves the lowest WER across all the tasks.
> >
> >
> > |**Method**| **Libri-10m** | **Libri-1h** | **Libri-10h** | **Dutch** | **Spanish** | **Mandarin** |
> > |:---:|:---:|:---:|:---:|:---:|:---:|:---:|
> > |Standard ft|38.40/44.98|18.96/26.30|9.70/18.19|19.82|13.86|26.67|
> > |Adaptor [2]|41.82/49.01|21.19/28.92|12.26/19.67|22.63|15.89 |29.03|
> > |**S3-Router**|**36.68/42.64**|**17.95/25.47**|**9.67/17.27**|**18.51**|**13.37**|**26.10**|
> >
> > In summary, we emphasize that while most existing speech adaptors aim to maintain the same performance as standard weight finetuning, our S3-Router often outperforms standard weight finetuning, e.g., a 1.72%/2.34% reduction in WER on LibriSpeech test-clean/test-other when being finetuned on LibriSpeech-10m  as shown in Section 4.2 of our manuscript.
> >
> > [1] “Efficient Adapter Transfer of Self-Supervised Speech Models for Automatic Speech Recognition”, B. Thomas et al., ICASSP 2022.
> >
> > [2] “Lightweight Adapter Tuning for Multilingual Speech Translation”, H. Le et al., ACL-IJCNLP 2021.
> >
> >
> > **3. Computational cost and real-device efficiency of our method:**
> >
> > **Our pruning results**: As clarified in our responses to Questions 1 and 2 above, our learned binary tasks are essentially unstructured sparsity and our method can match or surpass the pruning results of SOTA ASR pruning methods, including the range of your mentioned >70% sparsity ratio, according to Figures 4 and 5 in Section 5 of our submitted manuscript, as well as Section 3 in our supplementary material.
> >
> > **Comparison with adaptors**: Different from speech adaptors which mostly target training efficiency via reducing the numbers of trainable parameters while slightly increasing the inference cost, our method targets inference efficiency. For single-task inference, the weights indicated as zero by the binary mask can be simply set to zero, and thus the binary masks are no longer needed during inference.
> >
> > **Real-device support**: Note that all existing ASR pruning methods, including the SOTA method PARP, still adopt unstructured pruning. Although it may not be well supported by vanilla Pytorch, different sparse matrix libraries (e.g., Sputnik[3] for sparse GPU kernels of deep learning form Google, as well as cuSparse/IntelMKL from NVIDIA/Intel, respectively), sparse tensor compilers (e.g., TACO [4] and [5]), and sparse accelerators (e.g., [6][7]) are potential choices to exploit such unstructured pruning for real-device acceleration. For multilingual/multitask inference, the aforementioned customized compilers or accelerators are potential choices for efficiently loading the task-specific binary masks to fulfill the promise of real-device acceleration, which is one of our on-going works.
> >
> > [3] “Sparse GPU Kernels for Deep Learning”, T. Gale et al., SC 2020.
> >
> > [4] “The Tensor Algebra Compiler”, F. Kjolstad et al., OOPSLA 2017.
> >
> > [5] “A Sparse Iteration Space Transformation Framework for Sparse Tensor Algebra”, R. Senanayake et al., OOPSLA 2020.
> >
> > [6] “SNAP: An Efficient Sparse Neural Acceleration Processor for Unstructured Sparse Deep Neural Network Inference”, J. Zhang et al., JSSC 2020.
> >
> > [7] “SIGMA: A Sparse and Irregular GEMM Accelerator with Flexible Interconnects for DNN Training”, E. Qin et al., HPCA 2020.

---

> > > ### Author Response · Authors · 2022-08-02
> > > **Author Response to Reviewer 13fn (Part 3)**
> > >
> > > **4. Practical use of our method:**
> > >
> > > As clarified in our responses to Questions 1 and 3 above, our S3-Router can match or surpass the pruning results of SOTA ASR pruning methods across different sparsities, including your mentioned sparsity range of >70%. Furthermore, our method is validated to achieve better downstream performances than standard weight finetuning in low-resource ASR, and can serve as a tool to analyze what is encoded in speech SSL models, all of which we humbly clarify have demonstrated the practical use of our method as elaborated in our introduction.
> > >
> > > **5. Generalization to other types of speech SSL models:**
> > >
> > > Thanks for the suggestion! As we know, most of the advanced speech SSL paradigms after the era of wav2vec 2.0 adopt transformers as the model backbone, and thus are close to wav2vec 2.0 in terms of model structure. Additionally, following your suggestion, we further apply our S3-Router on the current SOTA speech SSL model data2vec (A. Baevski et al., ICML 2022) which features a new SSL pretraining paradigm as compared to wav2vec 2.0.
> > >
> > > As shown in the table below, we can observe that our S3-Router still achieves a better or comparable performance across all the datasets, including LibriSpeech-10m/1h/10h and dutch/spanish/mandarin from CommonVoice. For example, our S3-Router achieves a 1.01%/0.48%/0.35% reduction in PER on dutch/spanish/mandarin as compared with standard finetuning. Here we follow the same finetuning setting as wav2vec 2.0 which is also the default finetuning setting of data2vec in Fairseq.
> > >
> > > | **Method** | **Libri-10m** | **Libri-1h** | **Libri-10h** | **Method** | **Dutch** | **Spanish** | **Mandarin** |
> > > |:---:|:---:|:---:|:---:|:---:|:---:|:---:|:---:|
> > > | Standard ft | 30.75/34.612 | 14.15/19.61 | 7.28/13.11 | Standard ft | 19.65 | 13.80 | 25.48 |
> > > | 0.07@S3-Router | 30.78/35.17 | 14.09/19.72 | 7.56/13.39 | 0.08@S3-Router | 19.01 | 13.85 | 26.22 |
> > > | 0.08@S3-Router | 30.70/34.45 | 13.96/19.41 | 7.43/13.34 | 0.06@S3-Router | **18.64** | **13.32** | 25.73 |
> > > | 0.09@S3-Router | **29.86/34.09** | **13.92/19.43** | 7.23/13.25 | 0.04@S3-Router | 19.03 | 13.39 | **25.13** |
> > > | 0.10@S3-Router | 31.30/35.07 | 14.27/20.10 | **7.05/12.98** |  |  |  |  |
> > >
> > > *PS: Both results on LibriSpeech test-clean/other are reported. X@S3-Router indicates S3-Router with a sparsity of X.*
> > >
> > > Finally, we also adopt S3-Router-P for pruning data2vec-base (including both self-attention and FFN modules) on LibriSpeech-10h with a sparsity of 50%/60%/70% as shown in the table below. We can observe that our method can reach a sparsity of 60% with only 1.20% WER increase on LibriSpeech test-clean, which further indicates the generalization capability of our method to other speech SSL models.
> > >
> > > | **Method** | **test-clean** |
> > > |:---:|:---:|
> > > | Standard ft | 7.28 |
> > > | 0.5@S3-Router-P | 7.83 |
> > > | 0.6@S3-Router-P | 8.48 |
> > > | 0.7@S3-Router-P | 10.84 |
> > >
> > >
> > > **6. Clarification for results on SUPERB:**
> > >
> > > For SUPERB, the detailed learning rate settings for each task may vary and are not publically released. Hence, we follow the default training configurations in the official github repo s3prl for both our method and baselines for ensuring  an apple-to-apple comparison.
> > >
> > >
> > > **7. Evidence for the higher risk of overfitting on downstream speech tasks as compared to NLP tasks:**
> > >
> > > Thanks for the good question! This hypothesis is based on the evidence that the commonly adopted benchmark for ASR features a more stringent low-source setting, thus the overparameterized speech SSL models may be easier to overfit, as compared to NLP ones. For example, LibriSpeech-10m only contains 48 sentences while the CoLA dataset in the GLUE benchmark for NLP contains 9594 sentences for training and development, where the former is a more stringent low-source setting. We will clarify this in the final version.

---

> > > > ### Author Response · Authors · 2022-08-02
> > > > **Author Response to Reviewer 13fn (Part 4)**
> > > >
> > > > **8. Applying other regularization techniques for reducing overfitting:**
> > > >
> > > > Thanks for the question! We follow your suggestion to integrate more regularization techniques with standard weight finetuning, including **1)** L2 decay on model weights, **2)** different dropout ratios, **3)** smaller learning rates, and **4)** finetuning parts of the model (i.e., finetuning the weights of only the self-attn or FFN modules), where all four settings are on wav2vec2-base and LibriSpeech-10m/1h (both results on test-clean/other are reported) and benchmarked with our S3-Router. Note that these regularization techniques are mostly orthogonal to our method, but we still use the reported results of S3-Router in our manuscript for a fair comparison.
> > > >
> > > > As shown in the table below, note that the first data row in each table is the default configuration of standard weight finetuning. We can observe that **(1)** a better tuned L2 decay/dropout ratio/smaller learning rates can indeed further enhance the performance of standard weight finetuning under a more stringent low-resource setting, according to the larger improvements on LibriSpeech-10m than LibriSpeech-1h; **(2)** consistent with our observations in Table 5 of our manuscript, finetuning the weights of only the FFN  leads to a comparable/slightly better performance as compared to finetuning the whole model; and **(3)** our S3-Router without tuning the hyperparameters still achieves a better or comparable performance across all the settings, indicating that our method remains the most competitive for alleviating overfitting in a low-resource setting. We clarify that these regularization techniques can be combined with our S3-Router to further enhance the achievable performance, which is our future work.
> > > >
> > > > | **Weight Decay** | **Libri-10m** | **Libri-1h** | **Dropout Ratio** | **Libri-10m** | **Libri-1h** |
> > > > |:---:|:---:|:---:|:---:|:---:|:---:|
> > > > | 0 | 38.40/44.98 | 18.96/26.30 | 0.1 | 38.40/44.98 | 18.96/26.30 |
> > > > | 1e-5 | **37.76/44.73** | **18.88/26.63** | 0.15 | **36.99/43.93** | **18.37/26.33** |
> > > > | 1e-6 | 37.90/44.35 | 18.95/29.19 | 0.2 | 37.24/44.13 | 18.88/26.65 |
> > > >
> > > >
> > > > | **Trainable Weights** | **Libri-10m** | **Libri-1h** | **Init Learning Rate** | **Libri-10m** | **Libri-1h** |
> > > > |:---:|:---:|:---:|:---:|:---:|:---:|
> > > > | all | 38.40/44.98 | 18.96/26.30 | 5e-5 | 38.40/44.98 | 18.96/26.30 |
> > > > | self-attn only | 40.27/48.39 | 21.22/29.64 | 2e-5 | **37.59/44.03** | **18.32/25.68** |
> > > > | FFN only | **38.07/45.05** | **18.60/26.78** | 1e-5 | 38.322/44.06 | 18.65/25.36 |
> > > >
> > > > | **Method** | **Libri-10m** | **Libri-1h** |
> > > > |:---:|:---:|:---:|
> > > > | S3-Router | **36.89/42.88** | **17.95/25.47** |
> > > > |  |  |  |
> > > >
> > > > We hope that we have answered/addressed all your concerns and please let us know if you have further questions. Thank you!

---

> > ### Comment · Reviewer_13fn · 2022-08-07
> > **Further comments based on the authors' response**
> >
> > I appreciate the authors for the careful replies to my comments. From the author's response, I don't think I misunderstood the paper in any aspect, but I do think the authors' may have some misunderstanding of my comments (and thus not really have addressed all my comments and concerns as the authors claimed), which I will explain each of them below:
> >
> > 1. Regarding the results, I mainly commented on the key finding:
> >
> > "simply discarding no more than 10% of model weights via only finetuning model connections of speech SSL models can achieve better accuracy over standard weight finetuning on downstream speech processing tasks. "
> >
> > The current SSL models were trained to be generic for many tasks, fairly large and often trained with dropout enabled, we certainly expect the model to have some redundancy, especially when taking a specific task into account. What I feel disappointing is, that there is only a 10% redundancy in Wav2Vec 2.0, and such a small redundancy might not be useful in practice based on the current hardware design.
> >
> > 2. In my previous comment
> >
> > "Based on the current hardware, routing within sparse models (e.g. MoE models or Pathway models) would be more efficient than routing each individual connection within a densely-connected model unless the findings indicate way more connections can be discarded (e.g. more than 70%) -- this would allow using sparse rather than dense matrix structures and ops."
> >
> > I meant to discard a large number of connections and still maintain reasonably good accuracy. I don't think a WER of 12.7 (with 70% sparsity) is a good number of test-clean even with Librispeech-10h. Note that Wav2Vec2-base has 95M parameters, and 30% means 28.5M parameters.
> >
> > 3. I would like to explain a bit more about my comment based on the authors' reply
> >
> > "(2) Note that in terms of single-task finetuning, although the binary masks are learned by the binary optimization in Eq.(1) of our manuscript, during inference the weights indicated as zero by the binary mask can be simply set to zero, thus the inference process is the same with the original model (if such sparsity is not utilized by the hardware)."
> >
> > I think the proposed method can be used in two ways: 1) keeping the original Wav2Vec2 encoder and storing each set of binary masks as a task-dependent adaptor; 2) keeping a separate sparsified Wav2Vec2 encoder for each task separately. I understood the authors did not do anything special during the inference process in this paper. However, with a constraint of keeping better/the same accuracy as the finetuning method, this makes the finding of "sparsifying 10% parameters without hurting the performance" less useful (as it also needs to store which parameter can be sparsified etc.).
> >
> > In practice 1) might be more useful than 2) since the adapters can be stored separately in a more compressed form (e.g. bits), which might increase the practical value of the proposed method (especially since the authors discussed the limited resources of the on-device ASR). This was the reason why I tried to view it as an adaptor/adaptation method. However, even though, the number of parameters associated with each adapter is still very large (compared to other adaptors/adaptation methods), it certainly requires applying the mask every time a different task is processed, which increased the number of calculations.
> >
> > 4. In the adaption methods widely studied in the speech community, sometimes only a few thousand parameters need to be used to achieve good results (e.g. the LHUC method). Even in the newly added adaptor baseline compared by the authors (B. Thomas et al., 2022), it seems only a few million parameters are added, which is much more efficient than the proposed method. In general, I would recommend the authors compare with standard adaptor/adaptation methods that have been widely accepted by the community, such as LHUC.
> >
> > Overall, I would be happy to increase my evaluation if the paper can achieve one of these:
> > 1. Propose a completely novel method. However, this is not the case here, as the method was taken from an existing publication and only possibly first applied to speech, and therefore the novelty is limited.
> >
> > 2. Show some findings that are general enough for all speech SSL models. However, this is also not the case since the authors only examined Wav2Vec 2.0. I think the findings could be different for different types of speech SSL models, since the size of encoders, dropout rate, pre-training objectives, and pre-training task and data, might all make a substantial influence on sparsity-related findings.
> >
> > 3. Demonstrate the method have high practical values. This can be shown either high sparsity could be achieved by maintaining a good accuracy and then lead to new types of hardware, or only have a very small or no increase in the computation and storage cost. However, as I re-explained again, I don't think this is the case here.

---

> > > ### Author Response · Authors · 2022-08-07
> > > **Further Author Response to Reviewer 13fn**
> > >
> > > Thanks for your timely response and we've further addressed your concerns below.
> > >
> > > **1. Practical values of our work:**
> > >
> > > First of all, we humbly emphasize the practical value is that **our method has already outperformed 1) standard finetuning at low sparsities with better task accuracy and 2) SOTA pruning techniques at larger sparsities with higher efficiency.**
> > >
> > > We clarify again that **our method achieves the best pruning results as compared to existing SOTA pruning methods (e.g., PARP [88]) for speech SSL models**, as validated in the whole Section 5 of our manuscript, as well as the whole Section 3 of our supplementary material. As mentioned, our method achieves a WER of 12.70 (w/o LM) and 7.93 (w/ LM) on wav2vec2-base trained on LibriSpeech-10h at a sparsity of 70%, i.e., 95M*0.75=71.25M parameters are pruned out, **which has already outperformed all existing ASR pruning methods according to the rightmost sub-figure in Figure 4 of our manuscript.** Additionally, when sparsifying <=60% weights, our method can maintain <1 WER increase on wav2vec2-base/LibriSpeech-10h, again achieving the best pruning effectiveness across all existing methods.
> > >
> > > Indeed, pruning speech SSL models is non-trivial according to the analysis in PARP [88], which shows that directly applying OMP/IMP leads to inferior results as it's more challenging to identify the redundant neurons under low-resource settings. We kindly refer you to the "ASR pruning" sub-section in Section 2 of our manuscript for more literature in this direction.
> > >
> > > **2. Generality of our method to other speech SSL models:**
> > >
> > > **We have generalized our method to the latest SOTA speech SSL model data2vec (ICML 2022) in Part 3 (Question 5) of our previous response to you.** Note that data2vec features a totally different pretraining paradigm based on self-distillation and achieves the current SOTA performances on ASR. We've shown the consistent performance gains and pruning effectiveness of our method on top of data2vec in terms of both finetuning and pruning. Hence, **we have validated the generality of our method across **different** **(1)** pretraining paradigms, **(2)** pretraining settings (monolingual in wav2vec2/multilingual in xlsr), **(3)** encoder sizes (wav2vec2-base/large/xlsr), and **(4)** downstream tasks/languages.**
> > >
> > > **3. Further clarification about "there is only a 10% redundancy in Wav2Vec 2.0"**:
> > >
> > > We humbly clarify that this claim may not be accurate, as in the literature the measurement of redundancy is often combined with weight finetuning on top of a well-trained model. However, in our setting, **pretrained speech SSL models know nothing about the downstream tasks while the task-/language-specific information is encoded by properly setting some weights to zero**. Our key insight is that model sparsity can play a similar role as model weights, which is also recognized and valued by all the other two reviewers. Additionally, **our pruning results can help better understand the redundancy in speech SSL models like wav2vec 2.0**. More justifications about the value of our findings regarding the role of sparsity are provided in **Part 1 Question 1(1)** of our previous response to you. We hope all these clarifications can help you re-evaluate our work's insightful perspective.
> > >
> > > **4. Compare with other adapters:**
> > >
> > > We've already benchmarked with two recent speech adapters (one of which is the latest speech adapter design) for speech SSL models in Part 2 of our previous response, from which we can see that our method still outperforms the adapters. Note that we've already selected the baselines that are designed for **(1)** speech SSL models (which mostly adopt transformer-based structures after the era of wav2vec 2.0) and **(2)** ASR tasks or have open-sourced implementation for reference. The LHUC (TASLP 2016) you mentioned was not originally designed for transformers, let alone transformer-based SSL models, thus we didn't benchmark with it.
> > >
> > > **5. Novelty of our work:**
> > >
> > > We clarify that **(1)** our work is the first to finetune connections via binary masks in the speech domain, which is non-trivially different from the NLP domain, because low-resource ASR can suffer from a higher risk of overfitting due to a more limited number of labeled data, e.g., 48 sentences in LibriSpeech-10m vs. 9594 sentences in CoLA @ GLUE benchmark for NLP; **(2)** Our work features a novel mask initialization and is thoroughly demonstrated to be effective in downstream finetuning, multilingual/multitask processing, and pruning; and **(3)** our work is extended to analyze the language-wise similarity, which shows non-trivial correlation with human expertise in phonetics, and such analysis is unique to the speech domain and can enhance our understanding in what pretrained SSL models have encoded. **Hence, we can expect that this work could enhance the community’s understanding of speech SSL models and shed light on future innovations in better utilizing speech SSL models.**

---

> > > > ### Comment · Reviewer_13fn · 2022-08-08
> > > > **Increased my rating to acknowledge the efforts.**
> > > >
> > > > I have been aware of the development of the model parameter pruning, and also discussed the actual deployment of such methods with an expert to validate my thoughts. I still think the results achieved with S3-Router aren't competitive enough to be considered as a practical parameter pruning method compared to alternative methods (e.g. https://arxiv.org/pdf/2110.08352.pdf less than 10% WER regressions on Librispeech test-clean with 70% sparsity, whereas 20%-30% by S3-Router if my calculations are correct).
> > > >
> > > > The best perspective I can find is to consider the paper as an experimental study paper, and thus the importance of the findings becomes critical. I appreciate the efforts to add new experiments based on data2vec, but data2vec and wav2vec have the same first author and it's possible some critical settings are shared. I think WavLM is possibly more important to be studied as it takes the top places on the SUPERB leaderboard.
> > > >
> > > > Overall, I would like to acknowledge the efforts the authors spend to respond to my comments and improve the paper, and therefore increased my rating from 3 to 4. I still think a NeurIPS paper may need a bit more novelty, and possibly a clearer focus.

---

> > > > > ### Author Response · Authors · 2022-08-09
> > > > > **3nd Author Response to Reviewer 13fn**
> > > > >
> > > > > Thank you for increasing the rating and recognizing our efforts in the author response.
> > > > >
> > > > > First, we have to humbly remind you that **the setting in Omni-sparsity DNN (H. Yang et al., ICASSP 2022) you mentioned is fundamentally different from ours, as they trained their model on the whole labeled LibriSpeech-960h whereas our method targets a low-resource setting with <= 10h labeled speech**, thus the results are not comparable. As analyzed in PARP [88], it is more challenging to identify redundant neurons under low-resource settings, e.g., commonly adopted pruning methods OMP/IMP will lead to inferior results. **Therefore, for speech SSL models under a low resource setting, our method has already outperformed (1) standard finetuning at low sparsities with better task accuracy and (2) SOTA ASR pruning techniques at larger sparsities with higher efficiency.** It is well recognized that how to further compress speech SSL models to satisfy the requirement of actual deployment as you mentioned is still an open research question, and our method has pushed forward the achievable accuracy-efficiency frontier of compression towards this direction.
> > > > >
> > > > > Second, thanks for the suggestion of applying our method on WavLM. The reason that we didn’t benchmark on it as it has not been integrated into Fairseq, and thus we may not have sufficient time to finish the benchmarks during the rebuttal period. We would have certainly attempted to do so if you could explicitly point out WavLM in your earlier responses. We will try to add such comparisons in our final version. Meanwhile, we clarify that **the pretraining paradigm in data2vec has already been quite different from wav2vec 2.0, since in data2vec the original quantization module is discarded and all targets are provided by a new self-distillation mechanism with a newly introduced teacher model updated via an exponentially moving average.** Therefore, we have demonstrated the generality of method in terms of both finetuning and pruning as detailed in our 1st and 2nd run of responses, which will be included in our final version.
> > > > >
> > > > > Finally, regarding the novelty, as just commented by Reviewer 2h6A that **“I believe the paper is clear on the contributions it makes and the findings remain interesting”**. Furthermore, we humbly remind that our method has presented a coherent logical flow in our contributions, i.e., **we identify an interesting finding regarding the role of sparsity in speech SSL models, which can inspire better schemes for finetuning/pruning/understanding speech SSL models.** We will further highlight our key contributions to the community and include our discussions in our final version.

---

> > > > > > ### Comment · Reviewer_13fn · 2022-08-09
> > > > > > **Final comments**
> > > > > >
> > > > > > I would like to thank the authors for the careful response to my comments every time, and I acknowledge and respect the comments from other reviewers. However, I believe the paper review is a highly subjective procedure and it is important to make evaluations independently from my own perspective.
> > > > > >
> > > > > > It indeed took me some time to find out what could be the more useful improvements (such as WavLM), but I think it is the authors' responsibility to improve the paper and an anonymous reviewer can point out problems and try to make constructive suggestions. The fundamental reason lies in the fact that a general claim was made but the experiments were only with one category of models in the initial submission. Including the experiments and analysis for other categories of speech SSL models (as also suggested by reviewer 2h6A) would certainly considerably cause a reshape of the paper, and I don't even know if such a major revision should trigger another review. Constraining the title and writing to focus on Wav2Vec 2 is a much easier choice, but it would further reduce the value of the paper.
> > > > > >
> > > > > > I understand methods like Omni-sparsity use very different settings, but that is how much a pruning method needs to achieve if it can be useful in practice. I understand S3-Router achieved better performance than PARP, but PARP was an original method with sufficient novelty to support its acceptance. As novelty wasn't an advantage here, I tried my best to see if I could evaluate the paper mainly based on the importance of the findings or the practical values (even considering to use as adapters/adaptation). However, it is still unfortunate after a lot of thinking and struggling. To conclude, it still seems to me the major value of this paper is to be the first to adapt the method from [49] to speech, conduct limited but careful experiments, and demonstrate positive results (but I am impressed by the great efforts the authors spent in the response).
> > > > > >
> > > > > > I increased the rating to borderline accept since possibly I have reasons to accept outweigh reasons to reject. My evaluations are certainly biased, but the discussions and conclusions certainly clarify things. Hope these would be useful for the decision-makers.

---

> > > > > > > ### Author Response · Authors · 2022-08-09
> > > > > > > **Final Response to Reviewer 13nf**
> > > > > > >
> > > > > > > Thank you again for recognizing our efforts during the author response period, and we believe our discussions could further strengthen the quality of our work and make it more accessible to all audiences. Indeed, we recognize that “the paper review is a highly subjective procedure”, thus we have tried to scholarly clarify and justify the contributions and positions of our work in this field to 1) avoid misunderstandings and 2) learn different perspectives so that we can further improve our work.
> > > > > > >
> > > > > > > Regarding your comment that “Including the experiments and analysis for other categories of speech SSL models (as also suggested by reviewer 2h6A) would certainly considerably cause a reshape of the paper”, we understand that it is one of the most useful aspects in top tier conferences to enable authors improve their final version based on constructive comments from the reviewers, i.e., considering other categories of speech SSL models in this case.
> > > > > > >
> > > > > > > Overall, we have justified and validated the novelty/generality of our method in our previous responses and will further highlight them and include our discussions in our final version as promised.
> > > > > > >
> > > > > > > Finally, we appreciate all your time and efforts in reviewing our paper and discussing it with us. We certainly learn and grow our understanding in the process, which will help us to improve this work.

---

### Official Review · Reviewer_2h6A · 2022-07-13

**Rating:** 7
**Confidence:** 5
**Soundness:** 4 excellent
**Presentation:** 2 fair
**Contribution:** 3 good

**Summary:**

This paper proposes modifying the fine-tuning procedure of speech SSL models like wav2vec 2.0 by applying a binary masking based finetuning where each weight is either kept or masked to 0 instead of modifying the model weights directly. This follows a similar recently proposed technique in the NLP domain. The paper's main contributions is application to speech domain, demonstrating that this approach can work well for ASR as well as other speech tasks such as diarization. All of this is achieved by simply learning to mask about 10% of the weights, which leads to greater efficiency of storing task-specific models. The authors propose a new method to choose masks in which they randomly choose masks for each layer but initialize the weights for the masks based on the magnitude of the weights, and show that this outperforms random masking and magnitude based masking.

**Questions:**

Have you tried combining weight fine-tuning with your binary masking scheme? For example, you can fine-tune weights followed by learning binary masks, or learning binary masks followed by weight fine-tuning, or do it jointly.

**Limitations:**

None that I can see.

**Strengths And Weaknesses:**

Strengths:
- Interesting and impactful approach showing that a light weight fine-tuning can outperform full model fine-tuning for speech related tasks and lead to more efficient finetuned model storage (or alternatively, somewhat sparse models).
- Thorough evaluation with different model sizes, in both mono-lingual and multi-lingual scenarios and across multiple tasks.

Weaknesses:
- This is not a super novel work given that it mostly transfers findings from NLP, but nonetheless the demonstrated effectiveness of the approach for speech tasks and a better performing mask initialization technique should be of interest to the community.
- The presentation is overly dramatic - perhaps the authors should reduce the use of terms like "win-win" "excitingly" etc.

---

> ### Author Response · Authors · 2022-08-02
> **Author Response to Reviewer 2h6A**
>
> Thanks for recognizing the insightful findings of our work as an “interesting and impactful approach”. We have addressed your comments/concerns as follows:
>
> **1. Novelty of our work:**
>
> Thanks for recognizing our work as “be of interest to the community”. We further clarify that **(1)** as you mentioned, our work is the first to finetune connections via binary masks in the speech domain, which is non-trivially different from the NLP domain, because low-resource ASR may suffer from a higher risk of overfitting due to more limited number of labeled data, e.g., LibriSpeech-10m only contains 48 sentences while the CoLA dataset in GLUE benchmark for NLP contains 9594 sentences for training and development; **(2)** Our S3-Router features a novel mask initialization and is thoroughly demonstrated to be effective in downstream finetuning, multilingual/multitask processing, and pruning; and **(3)** our S3-Router is extended to analyze the language-wise similarity, which shows non-trivial correlation with human expertise in phonetics, and such analysis is unique to the speech domain and can enhance our understanding in what pretrained SSL models have encoded. Hence, we can expect that this work could enhance the community’s understanding of speech SSL models and shed light on future innovations in better utilizing speech SSL models.
>
> **2. Presentation style:**
>
> Thanks for your kind suggestion! We were hoping to highlight both the interestingness of our findings and their resulting benefits via our current presentation style for ease and joyful reading of readers. We will follow your suggestion to change the tone in our final version to improve our presentation to be more accessible to all audiences.
>
> **3. Combine weight fine-tuning with the binary masking scheme:**
>
> Thanks for the good question! Following your suggestion, we try the three suggested combinations on top of wav2vec2-base on LibriSpeech-10m/1h under a sparsity of 0.9 and the following three settings: **(1)** finetuning the model weights first and then learning binary masks, **(2)** learning binary masks first, which are then fused into model weights (i.e., set some weights to zero indicated by the learned masks), followed by weight finetuning, and **(3)** jointly finetuning both the weights and binary masks.
>
> Here we report the achievable WER (i.e., the lowest WER on test sets during the finetuning process) on both LibriSpeech test-clean/other in the table below. We can observe that **for experiment setting (1)** above, it achieves a comparable performance as learning binary masks only in terms of the achievable WER, which aligns with our hypothesis that SSL pretrained speech models have learned generalizable features and naturally contain sufficient information required by downstream tasks, thus weight finetuning may not be necessary for achieving a decent downstream performance; **For experiment setting (2)**, its achievable WER is comparable or slightly worse as compared to that of only learning masks and we observe that its achieved WER quickly reaches the lowest one in the first 1000 finetuning steps, indicating that the learned binary masks have already hit decent optimum; and **for experiment setting (3)**, we find that its achievable WER is higher than other combinations, indicating the joint optimization may lead to relatively poor convergence, which may be because the binary optimization on the masks as formulated in Eq.(1) of our manuscript favors more stable weight distributions as each weight is either masked out (i.e., set to zero) or fully kept during each training iteration.
>
>
> | **Training Set** | **Mask Only (S3-Router)** | **First-Weight-Then-Mask** | **First-Mask-Then-Weight** | **Joint Optimization** |
> |:---:|:---:|:---:|:---:|:---:|
> | LibriSpeech-10m | 36.89/42.88 | **36.79/42.92** | 37.31/43.49 | 40.84/47.44 |
> | LibriSpeech-1h | **17.95/25.47** | 18.01/25.55 | 18.32/25.97 | 19.74/28.83 |

---

> > ### Comment · Reviewer_2h6A · 2022-08-08
> > **Update**
> >
> > Thank you for your response and the additional experiments you've ran. I have increased my rating after your responses to my questions as well as to other reviewers. While I totally understand some of the concerns expressed by reviewer 13fn, from my perspective, you have addressed most of their major points, and what remains is a question of positioning. Personally, I believe the paper is clear on the contributions it makes and the findings remain interesting.
> >
> > Including data2vec results which you already have, and possibly other SSL models (e.g. HUBERT/WavLM/TERA) should also strengthen your claims.

---

> > > ### Author Response · Authors · 2022-08-08
> > > **Response to Reviewer 2h6A**
> > >
> > > Thank you very much for your constructive and insightful review and for recognizing the contributions of our work, and we will certainly include data2vec results as well as other SSL models in our final version. As we promised, we will open source all the codes and pretrained models upon acceptance.

---

### Official Review · Reviewer_nV4F · 2022-07-21

**Rating:** 7
**Confidence:** 4
**Soundness:** 2 fair
**Presentation:** 3 good
**Contribution:** 3 good

**Summary:**

This paper presents a novel fine-tuning method for downstream speech tasks, notably automatic speech recognition (ASR). The key idea of its method is to learn binary masks on the connectivities of pretrained weights instead of directly updating the weight values. The binary masks are obtained by discretizing learnable masks (which have the same shape as weights) during the forward pass. The masks can be optimized via stochastic gradient descent, where straight-through is used to approximate the  gradient estimation. The training objective contains a sparsity constraint, which also makes the proposed fine-tuning method to become a pruning technique.

**Questions:**

1. I am curious what kinds of regularization methods can be applied apart from the sparsity constraint. For example, has/have the author(s) try combining S3-router with dropout?

1-2. Can you add noise to the mask (dropping out the mask)?

2. The authors mentioned that S3-router can be seen as an all-in-one technique for on-device applications. Achieving such good results on multiple language tasks by only using less than 10% of the pretrained model for each task is impressive. However, if there are multiple language tasks, and if each (language | task)-specific mask vary a lot by each different task, then I wonder there will be a diminishing return in the pruning aspect. Because, in order to work well for a variety of tasks, the task specific mask should specialize (and should different from masks for other tasks), then this means each mask would be sparse, but the union set of sparse masks will not. Therefore, the whole pretrained model should be uploaded to the device. And as the authors pointed out the resource situation is quite limited. If a set of downstream tasks involve many different languages, I assume that the learned masks shouldn't be too close to each other, as the key idea of this new fine-tuning scheme and pruning technique is to learn (language | task)-specific mask. What is your opinion about this matter?

3. Can you share more details about training? There are no hyper-parameters, optimization details and training time in the paper.


**Limitations:**

The experimental results are strong, but at the same time, the proposed method is only demonstrated with empirical results. I am a bit wary of generalization capability of this method.


**Strengths And Weaknesses:**

Overall idea of the paper is simple but quite interesting. It's surprising to see how effective this method is, without changing pretrained weights directly, but only learning connectivity patterns. The experimental results look strong, so empirically the method seems to be working quite well. It would be nice if the authors can add a bit more refined explanation and analysis on why this method works well.

There are too many contents and citations, which hurt the readability. It is quite often distracting, because I have to open other papers to find out the details. It would be nice if the authors can make the paper to become more self-contained and make it simpler to read. For example, having to be forced to open a cited paper to check the size of the pretrained model (wav2vec) is a bit frustrating. Also, one of the most important details of a new method: "so how can I train this?" part is missing. The loss function is trivially given as an example, e.g., CTC loss. Having many experimental results on a paper is not a bad thing, but the training procedure and how the results were obtained are also important details. They shouldn't be omitted to have more space to add a few more tables.

As a reviewer I am pointing out that I am concerned about the reproducibility of this work as training details are not shared.

---

> ### Author Response · Authors · 2022-08-02
> **Author Response to Reviewer nV4F (Part 1)**
>
> Thanks for recognizing the insightful findings of our work as “simple but quite interesting” and “strong experimental results”. We have addressed your comments/concerns as follows:
>
> **1. Training hyperparameters and reproducibility:**
>
> **As promised in the abstract, we will certainly open source all our codes and pretrained models if being accepted. In fact, we provided the training hyperparameters in Section 5 of our submitted supplementary material.**  Basically, our codes are built on top of the open-source framework Fairseq and follow the officially provided optimizer (i.e., Adam), learning rate schedules (i.e., the tri-stage schedule proposed in wav2vec 2.0 with an initial learning rate of 5e-5), and other training hyperparameters in the default configuration files. Following your suggestion, we have added an description of our training settings in the revised manuscript, which will be included in the final version of our manuscript.
>
> **2. Readability of our contents and citations:**
>
> Thanks for your constructive suggestion! We aimed to elaborate more on our insights and experimental findings in our main content but we agree that adding more details and background can greatly enhance the readability of our paper. We will follow your suggestion to add more detailed description in our final version, while we have added the size of pretrained models and the training time in our revised manuscript.
>
> **3. More explanation and analysis on why the proposed method works well:**
>
> As analyzed in Section 3.1 of our submitted manuscript, our work is inspired by pioneering works which imply that tuning the connections of the model structure via introducing sparsity can play a similar role as finetuning the model weights. Here we attribute the success of our method to mainly two reasons: **(1)** SSL pretrained speech models have encoded generalizable features and naturally contained sufficient information required by downstream tasks, from which we only need proper ways to decode task-specific information. In our work, we find properly discarding some of the weights can effectively achieve this goal; and **(2)** considering that large-scale speech SSL models tend to suffer from overfitting under low-resource settings (according to [10, 11, 12, 13, 14] in our reference), our method of using a binary optimization process on the masks can potentially regularize the learning process and lead to better or comparable downstream performances, as compared to finetuning the overparameterized model weights, based on our empirical experiments across various models/datasets. We will add these discussions in the final version.
>
>
> **4. Regularization methods that can be applied apart from the sparsity constraint:**
>
> Thanks for your suggestion! First, we find that the L0 sparsity constraint is a key component for the success of our binary optimization as formulated in Eq.(1) of our  manuscript, which can do  more than merely a regularization effect. Since task-specific information is encoded with sparsity, it is necessary to indicate one weight is either masked out (i.e., set to zero) or fully kept during each training iteration for guaranteeing good convergence. If using an L1/L2 constraint on the learnable masks, there will exist a gap between training (non-binary masks) and inference (binary masks), which for example leads to >3% WER increase on LibriSpeech.
>
> For dropout, our reported results follow the default setting of wav2vec 2.0 in Fairseq to enable a dropout ratio of 0.1 after the self-attention and FFN modules, which is equivalent to what you suggested, i.e., applying dropout on masks. We find that S3-Router with dropout leads to a 0.33/0.68 WER reduction on LibriSpeech test-clean/test-other as compared to S3-Router w/o dropout (in particular, 36.68/42.64 vs. 37.01/43.32) under a sparsity of 8% on wav2vec2-base when training on LibriSpeech-10m, indicating that standard dropout is beneficial when combining with S3-Router. While regularization methods are orthogonal to our method, combining our method with other advanced regularization techniques could potentially further enhance the downstream performances.

---

> > ### Author Response · Authors · 2022-08-02
> > **Author Response to Reviewer nV4F (Part 2)**
> >
> > **5. A diminishing return in the pruning aspect under the multilingual setting:**
> >
> > Thanks for the excellent question! First, you are right that the learned masks for each language are not close to each other under a large sparsity ratio, thus your mentioned diminishing return in the pruning aspect under a multilingual/multitask setting can happen. However, we clarify that **(1)** under a multilingual/multitask setting, our method only needs to upload one copy of pretrained model while standard weight finetuning requires to upload N models for N languages/tasks, thus our method still wins the efficiency from this perspective; and **(2)** one way to tackle the diminishing return is to share the pruning patterns across languages/tasks in  early layers and only learn language-/task-specific masks on later layers of the model, leveraging the observation that task-specific information are mostly learned in later layers which is validated in Section 4/Table 2 of our supplementary material. We validate this on top of wav2vec2-base for dutch/mandarin/spanish. In particular, we enforce a shared sparsity of 60% on the first 6 transformer blocks, whose binary masks are transferred from the ones learned on LibriSpeech-10h, and learn language-specific masks with a sparsity of 60% on the last 6 transformer blocks. We can observe that no more than 0.9 PER increase as compared to independent pruning for each language under 60% sparsity (i.e., 23.02%/28.97%/16.13% vs. 22.17%/28.20%/15.33% for dutch/mandarin/spanish).
> >
> > We will discuss this aspect in the final version and clarify that our S3-Router can achieve decent finetuning performance and pruning effectiveness, while how to simultaneously and maximally win both is one of our future works.
> >
> >
> > **6. Generalization capability and theoretical analysis of our method:**
> >
> > We have provided more analysis about the success of our method in the aforementioned Question 3. In addition, we clarify that while it is still an open research question regarding the theoretical understanding of what is learned by speech SSL models, our method can serve as an alternative to help us understand what information pretained speech SSL models encode as shown in Section 6 of our manuscript.
> >
> > To further validate the generalization capability of our method, we apply our S3-Router on the current SOTA speech SSL model data2vec (A. Baevski et al., ICML 2022) which features a new SSL pretraining paradigm as compared to wav2vec 2.0. As shown in the table below, we can observe that our S3-Router still achieves better or comparable performances across all the datasets, including LibriSpeech-10m/1h/10h and dutch/spanish/mandarin from CommonVoice. For example, our S3-Router achieves a 1.01%/0.48%/0.35% reduction in PER on dutch/spanish/mandarin, as compared with standard finetuning. Here we follow the same finetuning setting as wav2vec 2.0 which is also the default finetuning setting of data2vec in Fairseq.
> >
> >
> > | **Method** | **Libri-10m** | **Libri-1h** | **Libri-10h** | **Method** | **Dutch** | **Spanish** | **Mandarin** |
> > |:---:|:---:|:---:|:---:|:---:|:---:|:---:|:---:|
> > | Standard ft | 30.75/34.612 | 14.15/19.61 | 7.28/13.11 | Standard ft | 19.65 | 13.80 | 25.48 |
> > | 0.07@S3-Router | 30.78/35.17 | 14.09/19.72 | 7.56/13.39 | 0.08@S3-Router | 19.01 | 13.85 | 26.22 |
> > | 0.08@S3-Router | 30.70/34.45 | 13.96/19.41 | 7.43/13.34 | 0.06@S3-Router | **18.64** | **13.32** | 25.73 |
> > | 0.09@S3-Router | **29.86/34.09** | **13.92/19.43** | 7.23/13.25 | 0.04@S3-Router | 19.03 | 13.39 | **25.13** |
> > | 0.10@S3-Router | 31.30/35.07 | 14.27/20.10 | **7.05/12.98** |  |  |  |  |
> >
> > *PS: Both results on LibriSpeech test-clean/other are reported. X@S3-Router indicates S3-Router with a sparsity of X.*
> >
> > Furthermore, we also adopt S3-Router-P for pruning data2vec-base (including both self-attention and FFN modules) on LibriSpeech-10h with a sparsity ratio of 50%/60%/70% as shown in the table below. We can observe that our method can reach a sparsity of 60% with only 1.20% WER increase on LibriSpeech test-clean, which further indicates the generalization capability of our method.
> >
> > | **Method** | **test-clean** |
> > |:---:|:---:|
> > | Standard ft | 7.28 |
> > | 0.5@S3-Router-P | 7.83 |
> > | 0.6@S3-Router-P | 8.48 |
> > | 0.7@S3-Router-P | 10.84 |

---

> > > ### Comment · Reviewer_nV4F · 2022-08-09
> > > **Update**
> > >
> > > I have read the response to my comments. I also read other reviewers' comments and again the response to those as well.
> > > I am more than happy to say that you have addressed most of my comments [1,2,3,4]. Thanks for taking my comments seriously.
> > >
> > > 5. I kind of agree with your point that although S3-Router can end up uploading the whole network when there are more languages/tasks, but other fine-tuning approaches might require uploading N networks. But if those approaches also use pruning, I am still wondering if the difference wouldn't be that huge.
> > >
> > > 6. I think the empirical results reported in this paper are fascinating already. Thanks for adding more experiments though! But when I think hard why the method works so well by simply learning the mask during fine-tuning, it's still a bit unclear to me after reading the paper. Also, 13fn pointed out, the paper may have weakness to position itself as a pruning method.
> > >
> > > That said, you have addressed the reviewers' comments as much as possible, and I feel the rebuttal process of this submission was constructive. My initial score was "weak accept" based on a few concerns that I had, and I left them as comments. Since you addressed the reproducibility bit and readability bit, I will raise my score to "accept" as half of my comments were resolved.

---

> > > > ### Author Response · Authors · 2022-08-10
> > > > **Final Author Response**
> > > >
> > > > Thank you for recognizing the contributions of our work and for your constructive comments, which can further strengthen the quality of our work. We will incorporate the discussed experiments and analysis to the final version. As we promised, we will open source all the codes upon acceptance.

---

### Meta-Review · Area_Chair_Zprw · 2022-09-09

**Recommendation:** Accept
**Confidence:** Certain

**Metareview:**

reviewers like that
* paper is well written and interesting
* experiments are solid

this paper should be accepted.

**Award:**

No

---

### Decision · Program_Chairs · 2022-09-14

Accept